# Well-Posed KL-Regularized Control via Wasserstein and Kalman–Wasserstein KL Divergences

Viktor Stein [1]  Adwait Datar [2]  Nihat Ay [2 3]

## Abstract

Kullback-Leibler (KL) divergence regularization is widely used in reinforcement learning, but it becomes infinite under support mismatch and can degenerate in low-noise regimes. Using a unified information-geometric framework, we introduce KL analogs by replacing the Fisher–Rao geometry in the dynamical formulation of the KL with transport-based geometries, and derive closed-form expressions for common distribution families. Between elliptic distributions, these divergences remain finite for degenerating equal covariances and yield a geometric interpretation of regularization heuristics used in Kalman ensemble methods. We demonstrate the utility of these divergences in KL-regularized optimal control. In the fully tractable setting of linear time-invariant systems with Gaussian process noise, the classical KL reduces to a quadratic control penalty that becomes singular as process noise vanishes. Our variants remove this singularity and yield well-posed problems. On a double integrator and a cart-pole example, the resulting controls preserve nontrivial feedback and achieve better closed-loop performance.

## 1. Introduction

The Kullback-Leibler (KL) divergence (Kullback & Leibler, 1951) between two probability measures $\mu, \nu \in \mathcal{P}(M)$,

$$\mathrm{KL}(\mu \mid \nu) := \begin{cases} \int_M \ln\left(\frac{\mathrm{d}\mu}{\mathrm{d}\nu}\right) \mathrm{d}\mu, & \text{if } \mu \ll \nu, \\ \infty, & \text{else} \end{cases} \quad (1)$$

can be interpreted in a dynamic, geometric way: if $\mu \ll \nu$, then

$$\mathrm{KL}(\mu \mid \nu) = \int_0^1 t \|\dot{\gamma}(t)\|_{\gamma(t)}^2 \, \mathrm{d}t,$$

where $\gamma$ is the Fisher-Rao dual geodesic with respect to the mixture connection with $\gamma(0) = \nu$ and $\gamma(1) = \mu$. (2)

Replacing the time weight $t$ in the integral by its average, $\frac{1}{2}$, results in a symmetric quantity, namely the squared geodesic distance induced by the Fisher–Rao metric. If we replace the Fisher-Rao geometry with the Wasserstein-2 geometry, this geodesic distance becomes the well-known Wasserstein-2 distance $W_2$ from optimal transport (Villani, 2003; 2009; Benamou & Brenier, 2000). This observation suggests a systematic way of constructing KL–type divergences: keep the formulation (2), but replace the underlying geometry.

The Fisher–Rao geometry is intrinsic to the statistical manifold, but it does not encode the geometry of the underlying state space $M$. Consequently, KL reacts discontinuously to support mismatch: for instance, $\mathrm{KL}(\delta_x \mid \delta_y) = \infty$ for all $x \neq y$, independently of the distance between $x$ and $y$. By contrast, the Wasserstein distance satisfies $W_2(\delta_x, \delta_y) = \|x - y\|_2$ and therefore retains information about the geometry of the sample space.

Motivated by this observation, we replace the Fisher–Rao geometry in (2) by Wasserstein, linearized Wasserstein, Kalman–Wasserstein, and Stein (with bilinear kernel) geometries and study the resulting divergence functionals. We refer to these functionals as *state-space-aware KL divergences*. They retain the asymmetric, action-based character of the KL divergence while incorporating geometric information from the state space.

KL regularization is ubiquitous in reinforcement learning and control. It appears in linearly-solvable and path-integral control formulations (Todorov, 2006; Kappen, 2005; Theodorou et al., 2010), in policy-search and trust-region methods (Peters et al., 2010; Schulman et al., 2015), and in recent optimal control and LQG/LQR-type formulations (Hashizume et al., 2024). However, the classical KL divergence can become problematic in low-noise regimes. In

[1]Department of Mathematics, Technical University of Munich & Munich Center for Machine Learning, Germany. The majority of the work was conducted while at the Institute of Mathematics at the Technical University of Berlin, Germany & the Berlin Mathematical School. [2]Institute for Data Science Foundations, Hamburg University of Technology, Germany [3]Santa Fe Institute, USA. Correspondence to: Viktor Stein <viktor.stein@tum.de>.

*Proceedings of the 43rd International Conference on Machine Learning*, Seoul, South Korea. PMLR 306, 2026. Copyright 2026 by the author(s).

particular, when controlled dynamics are compared to passive dynamics with small Gaussian process noise, the KL penalty reduces to a quadratic control cost whose coefficient scales inversely with the noise covariance. As the noise covariance vanishes, this penalty diverges, and the corresponding regularized control problem becomes singular. Related low-variance instabilities have also been observed in KL-regularized reinforcement learning from demonstrations (Rudner et al., 2021).

To isolate this pathology in a tractable setting, we study divergence-regularized control for LQRs with linear time-invariant dynamics and Gaussian process noise. In this regime, the classical KL penalty degenerates in the deterministic limit, while the Wasserstein KL and Kalman–Wasserstein KL penalties remain finite. Thus, the proposed divergences yield nontrivial optimal controls across both high-noise and low-noise regimes.

Our goal is not to propose a complete deep RL algorithm, but to develop a geometric framework for KL-type divergences obtained by replacing the Fisher–Rao geometry with state-space-aware geometries. For structured distribution classes, this yields closed-form formulas and tractable control objectives compatible with commonly used Gaussian policy parameterizations in continuous-control RL (Schulman et al., 2015).

### 1.1. Prior Work and Contributions

Our construction builds on the canonical contrast functions introduced in (Felice & Ay, 2021; Ay & Amari, 2015). The Wasserstein version of this contrast function was studied in (Ay, 2024), and explicit formulas for Gaussian measures were derived in (Datar & Ay, 2026a). The present paper extends this line of work by formulating a general construction, deriving explicit formulas for several geometries and distribution families.

The main contributions of this paper are as follows.

1. For a geometry $G$, we introduce $G$-dual geodesics and define the associated divergence $D^G$ as a time-weighted action integral along these curves. The (reverse) KL divergence is recovered when $G$ is the Fisher–Rao geometry, while Wasserstein-type choices of $G$ yield state-space-aware KL analogs.

2. We show that $G$-dual geodesics can be characterized as $G$-metric gradient flows of a potential energy. This links the proposed divergences to geometric optimization and to mean-field descriptions of neural-network dynamics (Chizat & Bach, 2020; Hardion & Lavenant, 2025).

3. We derive explicit formulas for $D^G$ between elliptic distributions in the Wasserstein, linearized Wasserstein,

and Kalman–Wasserstein geometries, and for centered Gaussians in the Stein geometry with a bilinear kernel. These formulas show precisely how transport-based geometries incorporate state-space information that is absent from the classical KL divergence.

4. We study divergence-regularized control in the LQR setting with Gaussian process noise. In this setting, the classical KL regularizer becomes singular as the noise covariance vanishes, whereas the Wasserstein KL and Kalman–Wasserstein KL remain finite and lead to well-posed optimal control problems.

5. We demonstrate on analytically tractable control examples that replacing KL regularization by the proposed state-space-aware divergences preserves key variational structure while avoiding the low-noise blow-up of the classical KL penalty.

For a comprehensive list of the symbols and notations used, see the appendix.

## 2. The Riemannian Structure of the Density Manifold

In this section, we review the infinite-dimensional geometry of the space of positive smooth probability densities (Lafferty, 1988; Bauer et al., 2016). We give examples such as the Fisher-Rao or Wasserstein metric, define a notion of geodesics, and show how these geodesics can be obtained by solving a gradient flow with respect to a linear functional.

The set of smooth positive probability densities on a compact $n$-dimensional Riemannian manifold $(M, \langle \cdot, \cdot \rangle, g)$ without boundaries equipped with its volume measure $\mu_g$,

$$P_+^\infty(M) := \left\{ \rho \in \mathcal{C}^\infty(M) : \rho > 0, \int_M \rho \, \mathrm{d}\mu_g = 1 \right\},$$

with the tangent space at $\rho \in P_+^\infty(M)$ being

$$T_\rho P_+^\infty(M) = \mathcal{S}_0^\infty(M)$$
$$:= \left\{ f \in \mathcal{C}^\infty(M), \int_M f \, \mathrm{d}\mu_g = 0 \right\},$$

and the cotangent space being defined as

$$T_\rho^* P_+^\infty(M) := \mathcal{C}^\infty(M)/\mathbb{R},$$

is a smooth infinite-dimensional Fréchet manifold.

As in (Ay & Schwachhöfer, 2026), we construct Riemannian metrics on $P_+^\infty(M)$ using Onsager operators.

**Definition 2.1** (Onsager operator). An *Onsager operator* (Liero & Mielke, 2013) (or raising of indices / musical

isomorphism) is specified by a collection of $L^2(M; \mu_g)$-selfadjoint and positive definite operators

$$\varphi_\rho^G \colon \mathcal{C}^\infty(M)/\mathbb{R} \to \mathcal{S}_0^\infty(M), \quad \rho \in P_+^\infty(M)$$

where $G$ denotes the geometry, e.g., Wasserstein. ◇

As in (Wang & Li, 2022), we can write a Riemannian metric on $P_+^\infty(M)$ associated to $\varphi^G$ either in terms of tangent vectors $a, b \in \mathcal{S}_0^\infty(M)$ or in terms of covectors of equivalence classes $[f], [g] \in \mathcal{C}^\infty(M)/\mathbb{R}$. They are related by $\varphi_\rho^G([f]) = a$ and $\varphi_\rho^G([g]) = b$.

**Definition 2.2** (Regular Riemannian metric on $P_+^\infty(M)$). The metric associated with an Onsager operator $\varphi^G$ is

$$\begin{aligned} g_\rho^G(a, b) &:= \langle a, (\Psi_\rho \circ (\varphi_\rho^G)^{-1})(b) \rangle_{L^2(M;\mu_g)} \\ &= \langle \varphi_\rho^G([f]), \Psi_\rho([g]) \rangle_{L^2(M;\mu_g)} =: \langle\!\langle [f], [g] \rangle\!\rangle_\rho^G, \end{aligned}$$

where $\Psi_\rho \colon \mathcal{C}^\infty(M)/\mathbb{R} \to \mathcal{C}_0^\infty(M, \rho)$, $[f] \mapsto f - \mathbb{E}_\rho[f]$ and $C_0^\infty(M; \rho) := \{ f \in C^\infty(M) : \langle f, \rho \rangle_{L^2(M;\mu_g)} = 0 \}$. The *density manifold* $P_+^\infty(M)$ equipped with this metric is denoted by $(P_+^\infty(M), g^G)$. ◇

We now detail examples of Riemannian metrics on $P_+^\infty(M)$.
*Example* 2.3 (Fisher-Rao metric). Let $\rho \in P_+^\infty(M)$. The Fisher-Rao Onsager operator is

$$\varphi_\rho^{\mathrm{FR}} \colon \mathcal{C}^\infty(M)/\mathbb{R} \to \mathcal{S}_0^\infty(M), \quad [f] \mapsto (f - \mathbb{E}_\rho[f])\rho.$$

Since the inverse of $\varphi_\rho^{\mathrm{FR}}$ has a closed form, we can write

$$g_\rho^{\mathrm{FR}}(a, b) := \int \frac{ab}{\rho} \, \mathrm{d}\mu_g.$$

The sectional curvature of $(P_+^\infty(M), g^{\mathrm{FR}})$ is $\equiv \frac{1}{4}$ (Friedrich, 1991). ●

*Example* 2.4 (Otto's Wasserstein metric with non-linear mobility). The Otto isomorphism (Otto, 2001) is given by the weighted Laplacian

$$\varphi_\rho^{\mathrm{O}} \colon \mathcal{C}^\infty(M)/\mathbb{R} \to \mathcal{S}_0^\infty(M), \ [f] \mapsto -\operatorname{div}_g(\rho \operatorname{grad}(f)).$$

The associated cotangent inner product is given by (using integration by parts)

$$\langle\!\langle [f], [g] \rangle\!\rangle_\rho^{\mathrm{O}} := \int_M \langle \operatorname{grad}(f), \operatorname{grad}(g) \rangle \rho \, \mathrm{d}\mu_g.$$

The curvature, Levi-Civita connection, and more of $(P_+^\infty(M), g^{\mathrm{O}})$ is extensively discussed in (Lott, 2008).

For a so-called *mobility function* $\mathrm{m} \colon [0, \infty) \to [0, \infty)$, modeling conductivity/permeability of the transport process, the Wasserstein-2 metric with nonlinear mobility (Carrillo et al., 2010; Dolbeault et al., 2009) has the Onsager operator

$$\begin{aligned} \varphi_\rho^{\mathrm{O_m}} \colon \mathcal{C}^\infty(M)/\mathbb{R} &\to \mathcal{S}_0^\infty(M), \\ [f] &\mapsto -\operatorname{div}_g(\mathrm{m}(\rho) \operatorname{grad}(f)), \end{aligned} \tag{3}$$

with associated cotangent inner product

$$\langle\!\langle [f], [g] \rangle\!\rangle_\rho^{\mathrm{O_m}} := \int_M \langle \operatorname{grad}(f), \mathrm{m}(\rho) \operatorname{grad}(g) \rangle \, \mathrm{d}\mu_g.$$

The Levi-Civita connection and the curvature of $(P_+^\infty, g^{\mathrm{O_m}})$ for $M = \mathbb{R}$ has been investigated in (Li, 2026).

A non-scalar-valued mobility arises in the Kalman-Wasserstein metric $g^{\mathrm{KW}}$ (Garbuno-Inigo et al., 2020) with mobility $\mathrm{m}_\lambda(f) := f(C(f) + \lambda \operatorname{id}_{TM})$, where $C(f)$ is the Riemannian covariance[1] of the density $f$, see (Abuqrais & Pigoli, 2026, Defn. 3.2), and $\lambda > 0$ is some regularization parameter.

For the constant mobility $\mathrm{m} \equiv 1$ we recover a flat $\dot{H}^{-1}$ ("linearized Wasserstein" (Peyre, 2018)) structure $g^{\dot{H}^{-1}}$. ●

## 2.1. Metric Gradient Flows on the Density Manifold

Consider a functional $E \colon P_+^\infty(M) \to \mathbb{R}$. We now define the covector field $\delta E$ on $P_+^\infty(M)$, induced by the ambient Fréchet space $\mathcal{C}^\infty(M)$.

**Definition 2.5** (Linear functional derivative). The *first variation* (or: linear functional derivative) of $E$, if it exists, is the one-form $\delta E \colon P_+^\infty(M) \to \mathcal{C}^\infty(M)/\mathbb{R}$ fulfilling

$$\langle f_\rho, \sigma \rangle_{L^2(M,\mu_g)} = \frac{\mathrm{d}}{\mathrm{d}t} E(\rho + t\sigma) \Big|_{t=0}, \quad [f_\rho] = \delta E(\rho).$$

for all $\sigma \in \mathcal{S}_0^\infty(M)$ such that $\rho + t\sigma \in P_+^\infty(M)$ for all sufficiently small $t \in \mathbb{R}$. ◇

For some examples of functional derivatives of common functionals, see (Stein & Li, 2025, Tab. 1).

We can then conveniently express the Riemannian gradient as the tangent vector field $\operatorname{grad}_\rho^G(E) := \varphi_\rho(\delta E(\rho))$.

In the following definition, we adopt a different sign convention than that of (Ay, 2024).

**Definition 2.6** (Metric gradient flow on $P_+^\infty(M)$). A curve $\rho \colon [0, \infty) \to P_+^\infty(M)$, $t \mapsto \rho_t$ is a $(P_+^\infty(M), G)$-*gradient flow of a functional* $E \colon P_+^\infty(M) \to \mathbb{R}$ *starting at* $\rho(0)$ if

$$\dot{\rho}(t) = -\operatorname{grad}_{\rho(t)}^G(E) \quad t > 0. \qquad ◇$$

We now generalize (Ay, 2024, Thm. 1).

**Proposition 2.7** (Metric gradients of the expectation functional). *For $f \in \mathcal{C}^\infty(M)$, we have*

$$\operatorname{grad}_\rho^G \mathbb{E}_{\cdot}[f] = \varphi_\rho^G([f]).$$

*These metrics are uniquely characterized by this property.*

*Proof.* For $F(\rho) := \mathbb{E}_\rho[f]$ we have $\delta F(\rho) = [f]$. The uniqueness follows exactly as in (Ay, 2024, Prop. 3). □

---

[1]For it to be well-defined, we require that on $M$, the 2-Fréchet mean is unique (Abuqrais & Pigoli, 2026, Subsec. 2.2), which holds, e.g., for Hadamard manifolds (Pennec, 2006).

## 2.2. Dual Geodesics on the Density Manifold

We now generalize the construction in (Felice & Ay, 2021), and define, analogously to (Ay, 2024, Eq. (33)) the *G-parallel transport*.

**Definition 2.8** (*G*-parallel transport). The *G-parallel transport* from $\rho \in P_+^\infty(M)$ to $\nu \in P_+^\infty(M)$ is

$$\Pi_{\rho,\nu}^G \colon T_\rho P_+^\infty(M) \to T_\nu P_+^\infty(M),$$
$$a \mapsto (\varphi_\nu^G \circ (\varphi_\rho^G)^{-1})(a). \qquad \diamond$$

Then, by construction of $g^G$ and $\Pi^G$, the resulting $G$ *connection* $\nabla^{(G)}$ is dual to the mixture connection $\nabla^{(m)}$ (Ay et al., 2017) in the sense of (Ay, 2024, Eq. (29)).

The geodesic $\gamma$, which is $G$-dual with respect to $\nabla^{(m)}$, or simply, the *G-dual geodesic* satisfying $\gamma(0) = \rho$ and $\dot\gamma(0) = a$ solves the equation

$$\dot\gamma(t) = \varphi_{\gamma(t)}^G \left( (\varphi_\rho^G)^{-1}(a) \right) \qquad (4)$$

on its interval of existence. We now make a precise connection between $G$-dual geodesics and the metric gradient flow of the expectation functional.

**Theorem 2.9** (Characterizing *G*-dual geodesics). *Let $\rho \in P_+^\infty(M)$ and $a \in S_0^\infty(M)$. For any $f \in \mathcal{C}^\infty(M)$ with $a = \varphi_\rho^G([f])$, the G-dual geodesic satisfying $\gamma(0) = \rho$ and $\dot\gamma(0) = a$ is the G-gradient flow of the* potential energy

$$\mathcal{F} \colon P_+^\infty(M) \to \mathbb{R}, \qquad \rho \mapsto -\mathbb{E}_\rho[f].$$

*Proof.* See Section A.1. □

# 3. State-Space-Aware KL Divergences

The main motivation for our canonical divergence is the geodesic projection property. Say that we have a point $p$ which we want to project onto a manifold $M$ via a divergence $D$, that is, we search for $p^* \in \arg\min_{q \in M} D(p \mid q)$. It is natural to assume that the geodesic connecting $p$ with $p^*$ meets $M$ orthogonally. This property is satisfied for (a) a Riemannian manifold where the divergence is given by half of the squared distance, the geodesic is given by the Levi-Civita connection, and the orthogonality is measured in terms of the Riemannian metric; (b) the probability simplex, where the divergence is given by the KL divergence or the $\alpha$-divergence, and the geodesics are the mixture geodesic or the corresponding $\alpha$-geodesic, respectively, and orthogonality is measured in terms of the Fisher-Rao metric. In information geometry, the geodesic projection property singles out particular divergences, which were studied as canonical divergences in (Ay & Amari, 2015; Felice & Ay, 2021) and are often asymmetric in their arguments. This projection property does not hold for many other commonly used divergences.

**Definition 3.1** (*G*-divergence). The *G*-divergence is

$$D^G \colon P_+^\infty(M) \times P_+^\infty(M) \to [0, \infty],$$
$$(\mu \mid \nu) \mapsto \inf \left\{ \int_0^1 t \, g_{\gamma(t)}^G(\dot\gamma(t), \dot\gamma(t)) \, \mathrm{d}t : \gamma \colon [0,1] \to P_+^\infty(M) \text{ is a } G\text{-dual geodesic}, \gamma(0) = \mu, \gamma(1) = \nu \right\}. \qquad (5) \; \diamond$$

*Example* 3.2. For $G = \mathrm{FR}$ we obtain the reverse KL divergence, see Theorem 4.9. For $G = \mathrm{O}$ we obtain the WKL divergence from (Datar & Ay, 2026a), see Theorem 4.10. ●

The following properties justify calling $D^G$ a divergence.

**Theorem 3.3** (Properties of $D^G$). *Let $\rho, \rho_1, \nu, \nu_1 \in P_+^\infty(M)$.*

1. *We have $D^G(\rho \mid \nu) \geq 0$. If there is a G-dual geodesic connecting $\rho$ and $\nu$, and it is unique or the infimum in (5) is attained, then $D^G(\rho \mid \nu) = 0$ if and only if $\rho = \nu$. However, $D^G(\rho \mid \nu)$ may take the value $+\infty$. In general, $D^G$ is not symmetric and does not fulfill the triangle inequality.*

2. *$D^G(\rho \otimes \rho_1 \mid \nu \otimes \nu_1) = D^G(\rho \mid \nu) + D^G(\rho_1 \mid \nu_1)$ for $G \in \{\mathrm{O}, \mathrm{KW}\}$ and if $\rho, \rho_1, \nu, \nu_1$ are Gaussians fulfilling the assumptions of Theorem 4.10 and Theorem 4.11, respectively.*

3. *On the set of elliptic distributions, the WKL is lower semicontinuous.*

4. *$D^G$ is invariant under isometries of the underlying space applied to both input measures simultaneously if the Onsager operator is equivariant under the action of that isometry.*

*Proof.* See Section A.2. □

*Remark* 3.4. The equivariance from Theorem 3.3 4. holds for any isometry $\Phi$ in the Fisher-Rao metric. It holds for the Otto metric (with non-linear mobility if it acts by scalar multiplication) as well.

*Remark* 3.5 (Energy dissipation interpretation). By Theorem 2.9, the *dissipation of $\mathcal{F}$ along $\gamma(t)$* is

$$\frac{\mathrm{d}}{\mathrm{d}t} \mathcal{F}(\gamma(t)) = \langle \Psi_{\gamma(t)}(\delta \mathcal{F}(\gamma(t))), \dot\gamma(t) \rangle_{L^2(M; \mu_g)}$$
$$\overset{(4)}{=} -\int_M \dot\gamma(t) \Psi_{\gamma(t)} \left( (\varphi_{\gamma(t)}^G)^{-1}(\dot\gamma(t)) \right) \mathrm{d}\mu_g$$
$$= -g_{\gamma(t)}^G(\dot\gamma(t), \dot\gamma(t)) \leq 0.$$

Hence, if there is a unique $G$-dual geodesic between $\rho$ and $\nu$, then the $G$-divergence between $\rho$ and $\nu$ can be conveniently

expressed as

$$D^G(\rho \mid \nu) = \int_0^1 t \left(-\frac{\mathrm{d}}{\mathrm{d}t} \mathcal{F}(\gamma(t))\right) \mathrm{d}t$$

$$= \int_0^1 \mathcal{F}(\gamma(t)) \, \mathrm{d}t - \mathcal{F}(\nu). \qquad (6)$$

## 4. Explicit Expressions for the State-Space-Aware KL Divergence

In the following, we consider the manifold $M = \mathbb{R}^d$, which is *non-compact* (see (Lafferty, 1988, pp. 730–731)). We will use the formal analogies of the formulas above, without claiming that they are always well-defined in this setting. [2]

First, we will treat the linearized Wasserstein metric $g^{\dot{H}^{-1}}$. Since this geometry is flat, the associated canonical divergence is symmetric and a squared norm.

**Theorem 4.1** (Linearized Wasserstein KL-divergence = Coulomb MMD). *For $\rho, \nu \in P_+^\infty(\mathbb{R}^d)$, the linearized Wasserstein divergence $D^{\dot{H}^{-1}}(\rho \mid \nu)$ is equal to*

$$\frac{1}{2} \int_{\mathbb{R}^d} \int_{\mathbb{R}^d} \Phi(x-y)(\rho(x)-\nu(x))(\rho(y)-\nu(y)) \, \mathrm{d}x \, \mathrm{d}y, \quad (7)$$

*where $\Phi$ is the fundamental solution of $-\Delta\Phi = \delta_0$.*

*Proof.* See Section A.3. □

*Remark* 4.2 (Wasserstein gradient flows of $D^{\dot{H}^{-1}}(\cdot \mid \nu)$). Up to the prefactor, the $\dot{H}^{-1}$ divergence (7) is the so-called maximum mean discrepancy (Borgwardt et al., 2006; Gretton et al., 2012) with the translation-invariant kernel $K(x,y) := \Phi(x-y)$. For $d = 1$, the kernel is $K(x,y) = -\frac{1}{2}|x-y|$ and the (regularized) Wasserstein gradient flow of $D^{\dot{H}^{-1}}$ is treated in (Duong et al., 2026; 2025). For higher dimensions, $\Phi$ is related to the Coulomb kernel, and the Wasserstein gradient flow of $D^{\dot{H}^{-1}}$ is discussed in (Boufadène & Vialard, 2025; Chizat et al., 2026).

### 4.1. The State-Space-Aware KL Divergence Between Elliptic Distributions

First, we recall elliptical distributions – the multivariate analogs of location-scale families (Schmidt, 2002, Sec. 5). This class is stable under affine pushforwards.

**Definition 4.3** (Elliptic distribution). A probability density $\rho \in P_+^\infty(\mathbb{R}^d)$ is called *elliptic* if there exist $m \in \mathbb{R}^d$, $\Sigma \in \mathrm{Sym}_+(\mathbb{R}; d)$ and a smooth, integrable function $g: [0,\infty) \to (0,\infty)$ with

- $\frac{\pi^{\frac{d}{2}}}{\Gamma\left(\frac{d}{2}\right)} \int_0^\infty r^{\frac{d}{2}-1} g(r) \, \mathrm{d}r = 1$ (normalization),

- $\int_0^\infty r^{\frac{d-1}{2}} g(r) \, \mathrm{d}r < \infty$ (finite expectation),

- $\int_0^\infty r^{\frac{d}{2}} g(r) \, \mathrm{d}r < \infty$ (finite variance),

such that

$$\rho = \rho_{m,\Sigma,g} = \det(\Sigma)^{-\frac{1}{2}} g\left((\cdot - m)^\mathsf{T} \Sigma^{-1}(\cdot - m)\right).$$

Then, we write $\rho \sim E(m, \Sigma, g)$. ◇

*Remark* 4.4 (Affine pushforward of elliptic distribution). Given the affine linear map $h(x) := Ax + b$ for some $A \in \mathbb{R}^{d \times d}$ and $b \in \mathbb{R}^d$, we have that $\rho \sim E(m, \Sigma, g)$ implies that $h_\# \rho \sim E(Am + b, A\Sigma A^\mathsf{T}, g)$ (Frahm et al., 2003, p. 279).

*Example* 4.5 (Elliptic distributions). For $g(x) = (2\pi)^{-\frac{d}{2}} e^{-\frac{1}{2}x}$ we obtain *Gaussian distributions* and for $g(x) = \frac{\Gamma\left(\frac{v+d}{2}\right)}{\Gamma\left(\frac{v}{2}\right)}(v\pi)^{-\frac{d}{2}} \left(1 + \frac{1}{v}x\right)^{-\frac{v+d}{2}}$ we obtain *Student's t distributions* with parameter $v > 2$. There are also other lesser-known examples, see (Schmidt, 2002, Sec. 6). ●

*Remark* 4.6. By Lemma B.1, a random variable $X$ with law $\rho \sim E(m, \Sigma, g)$ has $\mathbb{E}[X] = m$ and $\mathbb{V}[X] = \kappa_g \Sigma$, where $\kappa_g := \frac{1}{d} \int_{\mathbb{R}^d} \|y\|^2 g(\|y\|^2) \, \mathrm{d}y$. For Gaussian distributions, we have $\kappa_g = 1$ and for Student's $t$ with $v$ degrees of freedom, we have $\kappa_g = \frac{v}{v-2}$.

**Quadratic potentials** For the transport-based metrics, we can obtain a $G$-dual geodesic from $\rho$ to $\nu$ staying in the submanifold $E(\cdot, \cdot, g) \subset \mathcal{P}_+^\infty$, by choosing the quadratic potential function $f(x) = \frac{1}{2} x^\mathsf{T} B x + b^\mathsf{T} x$, where $b \in \mathbb{R}^d$, and $B \in \mathbb{R}^{d \times d}$ is symmetric. At present, we only conjecture that this is the unique geodesic, and we assume this conjecture holds in the following. We will find a closed-form expression for the gradient flow starting at $\rho$ and determine $B$ and $b$ in terms of $\rho$ and $\nu$.

The transport-based metrics have the *divergence form*

$$\varphi_\rho^G([f]) = -\nabla \cdot \left(\rho \cdot v^G(\rho, \nabla f)\right) \qquad (8)$$

for some *velocity field* $v^G: P_+^\infty(M) \times \mathcal{C}^\infty(M) \to \mathcal{C}^\infty(M)$.

*Example* 4.7 (Affine vector fields for elliptic distributions). For $\lambda > 0$, we have

$$v^{\mathrm{KW}}(\rho_{m,\Sigma,g}, \nabla f) = x \mapsto (\kappa_g \Sigma + \lambda I_d)(Bx + b), \quad (9)$$

as is verified in Section B.1. Similarly, $v^{\mathrm{O}}(\rho, \nabla f) = \nabla f$. These vector fields are affine linear functions. ●

**Lemma 4.8** (Elliptic closure for affine linear vector fields). *Suppose $\dot{\rho}_t = -\nabla \cdot (\rho_t v_t)$ for an affine vector field $v_t(x) := S_t x + s_t$ smoothly varying in $t$, with $S_t \in \mathbb{R}^{d \times d}$ and $s_t \in \mathbb{R}^d$ for all $t \geq 0$. If $\rho_0 \sim E(m_0, \Sigma_0, g)$, then $\rho_t \sim E(m_t, \Sigma_t, g)$, where $(m_t, \Sigma_t)$ solves*

$$\begin{cases} \dot{m}_t = S_t m_t + s_t, \\ \dot{\Sigma}_t = S_t \Sigma_t + \Sigma_t S_t^\mathsf{T} =: 2 \, \mathrm{Sym}(S_t \Sigma_t). \end{cases} \qquad (10)$$

---

[2]See the preprint (Datar & Ay, 2026b) for a treatment in the non-compact setting, specializing to Gaussian measures on $\mathbb{R}^n$.

*Proof.* See Section A.4. □

We will obtain explicit solutions of the system (10). Thus, by (6) and Lemma B.2 we can then evaluate the divergence as

$$
D^G(\rho \mid \nu) = \frac{\kappa_g}{2} \operatorname{tr}(B\Sigma_1) + \frac{1}{2} m_1^\mathsf{T}(Bm_1 + 2b)
$$
$$
- \int_0^1 \frac{\kappa_g}{2} \operatorname{tr}(B\Sigma_t) + \frac{1}{2} m_t^\mathsf{T}(Bm_t + 2b) \, \mathrm{d}t. \tag{11}
$$

First, we recall the following well-known result.

**Theorem 4.9.** *For $\rho, \nu \in P_+^\infty(\mathbb{R}^d)$ satisfying the assumption (30) in the Appendix we have $D^{\mathrm{FR}}(\rho \mid \nu) = \mathrm{KL}(\nu \mid \rho)$.*

*Proof.* See Section B.2. □

The next result provides a closed-form expression for the WKL divergence between two elliptic distributions and is a trivial extension of the result for Gaussian distributions developed in (Datar & Ay, 2026a).

**Theorem 4.10** (WKL divergence). *Let $\rho \sim E(m_0, \Sigma_0, g)$ and $\nu \sim E(m_1, \Sigma_1, g)$ and set $\Delta m = m_1 - m_0$. Then*

$$
D^O(\rho \mid \nu) = \frac{\kappa_g}{4} \operatorname{tr}\left(\Sigma_0 - \Sigma_1 + \Sigma_0 R^2 \log(R^2)\right)
$$
$$
+ \frac{1}{4} \left\| \sqrt{Q + 2\log(R)^\perp} \Delta m \right\|^2, \tag{12}
$$

*where $A^\perp := I - AA^\dagger$ for any symmetric matrix $A$, and*

$$
R = \Sigma_0^{-\frac{1}{2}} \left(\Sigma_0^{\frac{1}{2}} \Sigma_1 \Sigma_0^{\frac{1}{2}}\right)^{\frac{1}{2}} \Sigma_0^{-\frac{1}{2}}
$$
$$
Q = (R - I)^\dagger \left(\log(R^2)R^2 - R^2 + I\right)(R - I)^\dagger.
$$

*Furthermore, if $\Sigma_0 \Sigma_1 = \Sigma_1 \Sigma_0$, then we get the following simplification:*

$$
D^O(\rho \mid \nu) = \frac{\kappa_g}{4} \left\| \sqrt{Q}\left(\sqrt{\Sigma_1} - \sqrt{\Sigma_0}\right) \right\|_F^2
$$
$$
+ \frac{1}{4} \left\| \sqrt{Q + 2\log(R)^\perp} \Delta m \right\|^2.
$$

*Proof.* See Section A.5. □

**Theorem 4.11** (KWKL divergence). *Let $\rho \sim E(m_0, \Sigma_0, g)$ and $\nu \sim E(m_1, \Sigma_1, g)$, where $\Sigma_0$ and $\Sigma_1$ are simultaneously diagonalizable. With $\Delta m := m_1 - m_0$ and $\Delta \Sigma := \Sigma_1 - \Sigma_0$ we have*

$$
D^{\mathrm{KW}}(\rho \mid \nu) = \frac{\kappa_g}{4\lambda} \operatorname{tr}(\Xi) + \frac{1}{2} \Delta m^\mathsf{T}\left(\Delta\Sigma^\perp \Sigma_{0,\lambda}^{-1} \Delta\Sigma^\perp\right) \Delta m
$$
$$
+ \frac{1}{4\lambda} \Delta m^\mathsf{T} P \Xi P \left((\sqrt{\Sigma_1} - \sqrt{\Sigma_0})^\dagger\right)^2 \Delta m,
$$

*where $P := \Delta\Sigma(\Delta\Sigma)^\dagger$ and $\Sigma_{\ell,\lambda} := \kappa_g \Sigma_\ell + \lambda I$ for $\ell \in \{0,1\}$, and*

$$
\Xi := \Sigma_1 \ln\left(\Sigma_1 \Sigma_0^{-1}\right) + \frac{1}{\kappa_g} \Sigma_{1,\lambda} \ln\left(\Sigma_{0,\lambda} \Sigma_{1,\lambda}^{-1}\right).
$$

*Proof.* See Section A.6. □

Thus, for elliptic distributions with equal covariance, taking KW-geodesics in the dynamical formulation of the KL (2) corresponds to the "ad-hoc numerical technique called [additive] covariance inflation" (Takeda & Sakajo, 2024) (also called *enforcing a noise floor*) often used in the Kalman filter literature.

**Corollary 4.12** ($D^{\mathrm{KW}}$ enforces a noise floor). *If $\rho \sim E(m_0, \Sigma, g)$ and $\nu \sim E(m_1, \Sigma, g)$, then*

$$
D^{\mathrm{KW}}(\rho \mid \nu) = \frac{1}{2}(\Delta m)^\mathsf{T}(\kappa_g \Sigma + \lambda I)^{-1} \Delta m. \tag{13}
$$

*In particular, for $\rho \sim \mathcal{N}(m_0, \Sigma)$ and $\nu \sim \mathcal{N}(m_1, \Sigma)$*

$$
D^{\mathrm{KW}}(\rho \mid \nu) = \mathrm{KL}\left(\mathcal{N}(m_0, \Sigma + \lambda I), \mathcal{N}(m_1, \Sigma + \lambda I)\right).
$$

In particular, if $\Sigma$ degenerates, then $D^{\mathrm{KW}}$ is still well-defined, unlike KL.

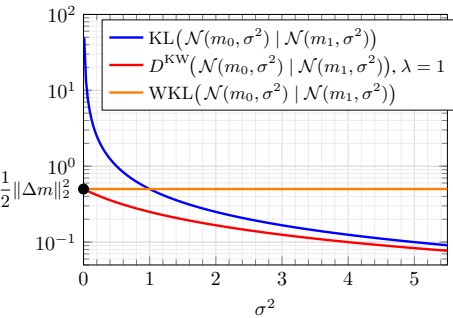

*Figure 1.* For $\rho \sim \mathcal{N}(m_0, \sigma^2 I_n)$ and $\nu \sim \mathcal{N}(m_1, \sigma^2 I_n)$, we compare $D^{\mathrm{KW}}(\rho \mid \nu) = \frac{1}{2(\sigma^2 + \lambda)} \|\Delta m\|_2^2$ with $\mathrm{KL}(\rho \mid \nu) = \frac{1}{2\sigma^2} \|\Delta m\|_2^2$ and $\mathrm{WKL}(\rho \mid \nu) = \frac{1}{2} \|\Delta m\|_2^2$, where $\lambda = 1$ and $\|\Delta m\|_2^2 = \frac{1}{4}$.

*Remark* 4.13 ($D^{\mathrm{KW}}$ interpolates between KL and WKL). In Figure 1 we observe that $D^{\mathrm{KW}}$ is a smoothing of KL, since $\lim_{\lambda \searrow 0} D^{\mathrm{KW}} = \mathrm{KL}$. For $\lambda = 1$ we observe that $D^{\mathrm{KW}}$ interpolates between WKL and KL, since then

$$
\lim_{\sigma^2 \to \infty} \frac{D^{\mathrm{KW}}(\rho \mid \nu)}{\mathrm{KL}(\rho \mid \nu)} = 1 = \lim_{\sigma^2 \searrow 0} \frac{D^{\mathrm{KW}}(\rho \mid \nu)}{\mathrm{WKL}(\rho \mid \nu)}.
$$

On the other hand, $D^{\mathrm{KW}}$ is non-local.

# 5. Optimal Control Problems with Divergence-Based Regularization

In this section, we study Wasserstein-based KL analogs (WKL and KWKL) as regularizers for stochastic linear–quadratic control. In classical KL-regularized control and path-integral formulations (Todorov, 2006; 2009; Kappen, 2005), the quadratic control cost scales with the inverse process noise covariance and becomes singular in the deterministic limit. By contrast, the WKL and KWKL penalties remain finite, yielding well-posed control problems and meaningful low-noise limits.

Consider linear time-invariant dynamics described by

$$x_{t+1} = Ax_t + Bu_t + w_t, \qquad t \in \mathbb{N}, \qquad (14)$$

where $x_t \in \mathbb{R}^n$ denotes the state, $u_t \in \mathbb{R}^m$ denotes the control input and $w_t \in \mathbb{R}^n$ denotes the process noise. Let $x_0 \sim \mathcal{N}(0, \Sigma_0)$ and assume the process noise $w_t$ consists of independent and identically distributed random variables distributed[3] according to $\mathcal{N}(0, \Sigma_w)$. We restrict attention to linear feedback policies $u_t = Fx_t$ and want to find the optimal gain matrix $F$ by minimizing the infinite-horizon discounted quadratic state cost

$$J(F) := \mathbb{E}_{w_t \sim \mathcal{N}(0, \Sigma_w)} \left[ \sum_{t=0}^{\infty} \frac{1}{2} \gamma^t x_t^\mathsf{T} Q x_t \right],$$

with *discount factor* $\gamma \in (0, 1)$ and a user-specified $Q \in \mathrm{Sym}_+(\mathbb{R}; n)$. To regularize the control effort, we penalize deviations between the controller's conditional state-transition distribution and the uncontrolled conditional state-transition distribution. Specifically, we add a divergence-based regularization term comparing

$$p_t := p(x_{t+1} \mid x_t, u_t) = \mathcal{N}(Ax_t + Bu_t, \Sigma_w),$$
$$p_0 := p(x_{t+1} \mid x_t, u_t = 0) = \mathcal{N}(Ax_t, \Sigma_w).$$

Computing the divergences yields $D^\circ(p_t \mid p_0) = \frac{1}{2} u_t^\mathsf{T} R_\circ u_t$, where

$$R_{\mathrm{FR}} = B^\mathsf{T} \Sigma_w^{-1} B, \qquad R_{\mathrm{O}} = B^\mathsf{T} B,$$
$$R_{\mathrm{KW}} = B^\mathsf{T} (\Sigma_w + \lambda I)^{-1} B. \qquad (15)$$

The KL-based regularizer explicitly depends on $\Sigma_w^{-1}$ and thus diverges as $\Sigma_w$ approaches singularity. In contrast, the KWKL-based and the WKL-based regularizers remain well-defined in this limit. With these regularization terms added to the state cost, the total cost with divergence-based regularization is

$$J^\circ(F) := \mathbb{E}_{w_t \sim \mathcal{N}(0, \Sigma_w)} \left[ \sum_{t=0}^{\infty} \gamma^t \Big( \frac{1}{2} x_t^\mathsf{T} Q x_t + D^\circ(p_t \mid p_0) \Big) \right],$$

---

[3]As demonstrated above, we could instead choose $w_t \sim E(0, \Sigma_w, g)$ and everything would work analogously.

where $\circ \in \{\mathrm{FR}, \mathrm{O}, \mathrm{KW}\}$. Since the divergence contributes an additive quadratic term in $u_t$, each formulation reduces to a discounted LQR problem with effective control penalties (15). The problems $\arg\min_F J^\circ(F)$ possess closed-form solutions in the form of solutions to Riccati equations (see Appendix C).

As a consequence of Theorem C.1, we obtain:

**Corollary 5.1** (Optimal policies). *Consider the LTI system (14) where $(A, B)$ is stabilizable, $(A, Q^{1/2})$ is detectable, and $B$ has full column rank. Let $\lambda > 0$ with $R_\circ$ as defined in (15). For each $\circ \in \{\mathrm{FR}, \mathrm{O}, \mathrm{KW}\}$, let $P_\circ \in \mathrm{Sym}_+(\mathbb{R}; n)$ denote the unique solution of the discrete algebraic Riccati equation (39) with the corresponding $R_\circ$. The optimal linear policy minimizing $J^\circ$ with $\gamma \in (0, 1)$ is*

$$F_\circ = -\gamma (R_\circ + \gamma B^\mathsf{T} P_\circ B)^{-1} B^\mathsf{T} P_\circ A. \qquad (16)$$

*In particular, $F_{\mathrm{O}}$ does not depend on $\Sigma_w$. If $\Sigma_w = \rho I$ for some $\rho > 0$ and $\lambda = 1$, then*

1. $\lim_{\rho \searrow 0} F_{\mathrm{KW}} = F_{\mathrm{O}}$.

2. $\lim_{\rho \to \infty} \|F_{\mathrm{KW}} - F_{\mathrm{FR}}\| = 0$.

3. *Additionally, if the spectral radius of $A$ is less than $\frac{1}{\sqrt{\gamma}}$, then $\lim_{\rho \searrow 0} F_{\mathrm{FR}} = 0$ while $\lim_{\rho \searrow 0} F_{\mathrm{KW}} = F_{\mathrm{O}}$ is not the zero policy in general.*

*Proof.* See Appendix C.1. $\square$

*Remark* 5.2. It can be shown that if $A$ has eigenvalues with magnitude greater than $\frac{1}{\sqrt{\gamma}}$, then $\lim_{\rho \to 0} F_{\mathrm{FR}}$ does not equal the zero policy. However, if $\alpha < \frac{1}{\sqrt{\gamma}}$ is an eigenvalue of $A$, then $\alpha$ is also an eigenvalue of the closed-loop system. Thus, if the open-loop system has slowly decaying modes, these are not stabilized under $F_{\mathrm{FR}}$, yielding degraded closed-loop performance.

## 5.1. Optimal control with path-integral formulation

Although our previous result was derived in the LMDP framework (Todorov, 2006; 2009; Dvijotham & Todorov, 2010; Guan et al., 2014), where control costs are stagewise divergence penalties, the same phenomenon also appears in the path-integral formulation of stochastic optimal control (Kappen, 2005; Theodorou et al., 2010).[4] In the special case of an identity input matrix, the low-noise degeneracy of path-integral KL regularization already follows from

---

[4]LMDP and path-integral formulations coincide for KL regularization, but not necessarily for our divergences, since they do not generally admit a chain-rule decomposition into stagewise costs; see Appendix C.2.

(Patil et al., 2024). To show that the behavior in Corollary 5.1 is not an artifact of stagewise decomposition, we study a linear–Gaussian finite-horizon setting and show that trajectory-level KL regularization exhibits the same low-noise pathology, while WKL and KWKL remain well posed in the low-noise limit.

We again consider (14), but now assume a fixed initial state $x_0 \in \mathbb{R}^n$. For a *finite* horizon of length $T$, stack the states, controls, and noises as $x, u, w$. Then, $x = \mathcal{A}x_0 + \mathcal{B}_u u + \mathcal{B}_w w$, where the matrices $\mathcal{A}$, $\mathcal{B}_u$ and $\mathcal{B}_w$ are given in Appendix C.3. The uncontrolled trajectory is obtained by setting $u = 0$, namely $x_{uc} = \mathcal{A}x_0 + \mathcal{B}_w w$. Consider the finite-horizon objective

$$V^\circ(u) = \mathbb{E}_{w_t \sim \mathcal{N}(0, \Sigma_w)} \left[ \tfrac{1}{2} x^\top \mathcal{Q} x \right] + D^\circ \Big( p(x) \mid p(x_{uc}) \Big).$$

This formulation naturally arises in model predictive control (MPC), where such finite-horizon problems are solved repeatedly in a receding-horizon.

In contrast to the stagewise regularization studied in the previous subsection, we now consider regularization defined through divergences between the controlled and uncontrolled trajectory distributions. Since $w \sim \mathcal{N}(0, I \otimes \Sigma_w)$, it is straightforward to verify that $D^\circ \Big( p(x) \mid p(x_{uc}) \Big) = \tfrac{1}{2} u^\mathsf{T} \mathcal{R}_\circ u$ for appropriate matrices $\mathcal{R}_\circ$ give in Appendix C.4. Thus, the divergences contribute an additive quadratic term in $u$, and each regularizer leads to a finite-dimensional convex quadratic program that can be solved in closed form. In this setting, it is possible to prove a result analogous to Corollary 5.1, which is informally stated next, deferring the formal result to Appendix C.5.

**Theorem 5.3** (Informal version of Optimal control under path-integral regularization)**.** *Let $u_\circ$ denote the optimal control sequence minimizing $V^\circ$ for $\circ \in \{\mathrm{FR}, \mathrm{O}, \mathrm{KW}\}$. When $\Sigma_w = \rho I$, $u_{\mathrm{FR}} \xrightarrow{\rho \to 0} 0$ while $u_{\mathrm{KW}} \xrightarrow{\rho \to 0} u_\mathrm{O}$ is non-zero in general. Further, $u_{\mathrm{KW}} \xrightarrow{\rho \to \infty} u_{\mathrm{FR}}$.*

### 5.2. Example 1: Double integrator

As a concrete example, we consider the one-dimensional double integrator system[5] (Recht, 2019) described by

$$\begin{bmatrix} q_{t+1} \\ p_{t+1} \end{bmatrix} = \begin{bmatrix} 1 & 1 \\ 0 & 1 \end{bmatrix} \begin{bmatrix} q_t \\ p_t \end{bmatrix} + \begin{bmatrix} 0 \\ 1 \end{bmatrix} u_t + w_t,$$

where $q_t$ is displacement, $p_t$ is velocity, $u_t$ is force and $w_t \sim \mathcal{N}(0, \rho I)$ with variance parameter $\rho \geq 0$. We set $Q = I$ and $\gamma = 0.9$. The assumptions of Corollary 5.1 hold.

As $\rho \searrow 0$, KL-regularization makes $J^{\mathrm{FR}}$ dominated by the divergence term, driving $F_{\mathrm{FR}} \to 0$ and yielding poor closed-loop performance. Figure 2 (and Figure 9) shows

---

[5] A double integrator is a canonical and simplified model of a mobile robot, such as a quadrotor.

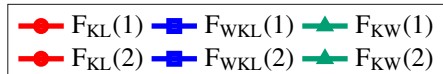

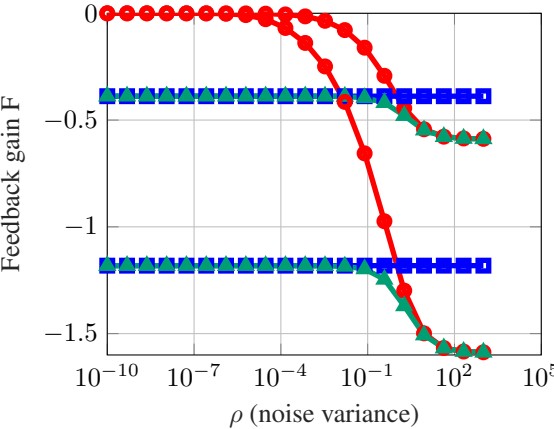

*Figure 2.* Optimal feedback gains for FR-, O-, and KW- regularized LQR as function $\rho$. Each feedback gain $F \in \mathbb{R}^{1 \times 2}$ shown component-wise; both entries are shown. KL gains shrink to zero as noise disappears, WKL gains are constant because they do not depend on $\rho$, and KW gains smoothly interpolate between the two.

$F_{\mathrm{FR}} \to 0$ while $F_\mathrm{O}$ and $\mathcal{F}_{\mathrm{KW}}$ remain nonzero ; for $\lambda = 1$, KW interpolates between WKL (low noise) and KL (high noise), consistent with Corollary 5.1 [6].

Closed-loop trajectories in Figure 3 illustrate the qualitative consequences: for small $\rho$, KL yields large oscillations due to vanishing feedback, whereas WKL and KW produce increasingly well-damped responses. To assess stability more directly, Figure 4 plots the spectral radius of $A + BF_\circ$ versus $\rho$. The KL-based closed-loop approaches the unit circle (stability boundary) as $\rho \to 0$, whereas WKL and KW avoid this pathology; KW again interpolates smoothly between the two regimes as $\lambda$ varies from 1 to 0. The optimal costs are plotted in Figure 7 in the Appendix C.8, which confirms these observations further.

### 5.3. Example 2: Cart-pole system

We consider the cart–pole system, a standard nonlinear control benchmark, and linearize it around the upright equilibrium at the origin. An LQR controller is designed using the linearized model. The resulting controller is applied to the full nonlinear dynamics to evaluate performance beyond the linear approximation (see Appendix C.6 for details). As with the double integrator, KL-regularization produces large oscillations, while WKL and KW yield increasingly stable trajectories as $\rho$ decreases, resulting in superior closed-loop performance. This is illustrated in Figure 6 which shows closed-loop trajectories of the nonlinear cart–pole under the

---

[6] The code is available at this GitHub repository.

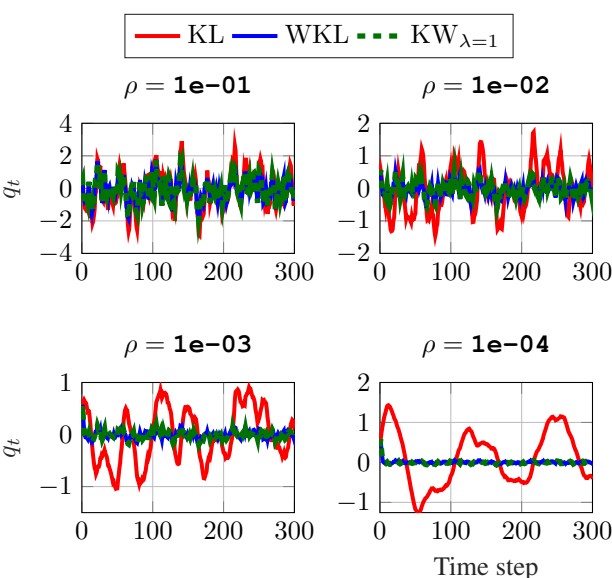

Figure 3. Closed-loop trajectories of the double integrator under KL-, WKL-, and KW-regularized controllers for $\rho \in \{10^{-1}, 10^{-2}, 10^{-3}, 10^{-4}\}$. For small $\rho$, KL yields weak feedback and large oscillations, whereas WKL- and KW-regularized controllers lead to reduced oscillations. See Figure 8 for other values of $\lambda$.

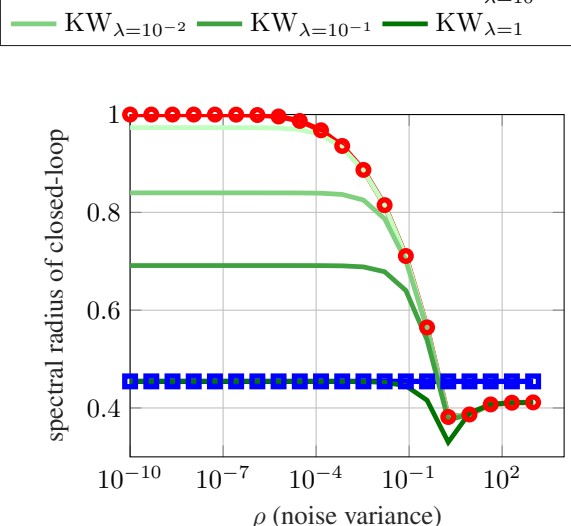

Figure 4. Closed-loop spectral radius as a function of noise $\rho$ for different values of parameter $\lambda$. KL-regularized controllers approach the unit circle as $\rho \to 0$, indicating near-instability, while WKL- and KW-regularized controllers maintain spectral radii well below one across noise regimes. Varying $\lambda$ interpolates between KL-like and WKL-like behavior.

LQR controller designed from the linearized model.

## 5.4. Example 3: Double integrator with anisotropic noise

We consider a double-integrator example with anisotropic process noise; the full experimental setup and the corresponding optimal feedback gains are given in Appendix C.7. The resulting LQR controllers exhibit a clear directional distinction: KL suppresses control in the low-noise dimension, WKL treats both dimensions identically, and KWKL adapts direction-wise, matching WKL in the low-noise direction and KL in the high-noise direction.

*Remark* 5.4 (Other regularizers). Many other divergences could be used to penalize the discrepancy between $p_t$ and $p_0$. However, even if some (like Jensen-Shannon divergence, squared Hellinger, or total variation, or maximum mean discrepancy) are bounded and thus not blow up for vanishing noise, or are explicitly modified to remain finite for measures with disjoint support (Glaser et al., 2021; Stein et al., 2026b), they either admit closed forms between elliptic distributions that are not quadratic (Nielsen & Okamura, 2024) or do not admit closed form expressions at all. For non-quadratic regularizers, the optimal policy $F$ is not available in closed form.

*Remark* 5.5. While the state-cost $Q$ remains problem-specific, our divergence-based regularization can be interpreted as a principled approach to selecting the control penalty matrix $R$ in LQR design. Whereas in classical control, $R$ is typically chosen heuristically, our framework determines an anisotropic control metric that depends on the noise covariance and the input matrix $B$, thereby fixing the relative penalty across the control directions.

## 6. Conclusions and Future Work

Motivated by certain shortcomings of the KL divergence as a regularizer, we introduced state-space-aware KL divergences by considering different geometries in the dynamical formulation of the KL. These novel divergences avoid the low-noise degeneracy of KL and yield improved closed-loop behavior, effortlessly carrying over to other noise models beyond Gaussians. Having validated our divergences in the fully tractable LQR setting, a natural next step is to integrate them into reinforcement-learning algorithms and empirically evaluate their behavior and benefits in both model-free and model-based RL.

**Acknowledgments.** The authors thank the reviewers for their comments, which improved this submission. VS thanks his advisor, Gabriele Steidl, for her support and guidance throughout, as well as the organizers of the *Conference on Mathematics of Machine Learning 2025* in Hamburg. The authors acknowledge fruitful conversations with Lorenz Schwachhöfer. AD and NA thank Takeru Matsuda for insightful discussions, particularly for suggesting extending the WKL divergence formulae to elliptic distributions. NA thanks Felix Otto for his valuable advice.

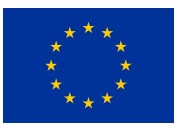 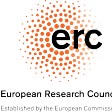

Funded by the European Union. Views and opinions expressed are, however, those of the authors only and do not necessarily reflect those of the European Union or the European Research Council Executive Agency. Neither the European Union nor the granting authority can be held responsible for them. This project has received funding from the European Research Council (ERC) under the European Union's Horizon Europe research and innovation programme (grant agreement No. 101198055, project acronym NEITALG).

## Impact Statement

This paper presents work whose goal is to advance the field of Machine Learning. There are many potential societal consequences of our work, none of which we feel must be specifically highlighted here.

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

# Appendix

**Notation**    The $n \times n$ identity matrix is denoted by $I_n$ and the set of symmetric positive definite matrices $\mathrm{Sym}_+(\mathbb{R}; d)$. The Moore-Penrose inverse of a matrix $A$ is denoted by $A^\dagger$. The orthogonal onto the kernel is $A^\perp := I - AA^\dagger$. The symmetric part of a matrix $A$ is denoted by $\mathrm{Sym}(A) := \frac{1}{2}(A + A^\top)$.

We denote the time derivative by a dot, like $\dot{\varphi}_t$. For a Borel measurable map $f \colon M \to Y$ between metric spaces and a probability measure $\mu = \rho\mu_g$ on $M$, its pushforward by $f$ is a probability measure on $Y$, denoted by $f_\#\mu$ and defined by $(f_\#\mu)(A) := \mu(f^{-1}(A))$ for any measurable set $A \subset Y$. We do not distinguish a probability measure from its density. We set $\mathbb{E}_\rho[f] := \int_M f\rho \, \mathrm{d}\mu_g$. By $\mathrm{grad}$ we denote the Riemannian gradient and by $\mathrm{div}_g$ the Riemannian divergence with respect to the volume measure.

**Structure of the appendix**    First, in Section A we present proofs for Section 4, and in Section B we perform simple but cumbersome calculations. Additionally, we define the Stein metric and compute the associated KL divergence for centered elliptic distributions. Then, in Section C, we give some background on control theory and prove the related theorems mentioned in the experiments section.

## A. Proofs

### A.1. Proof of Theorem 2.9

### A.2. Proof of Theorem 3.3

*Proof.*    1. Since the infimum is taken over a nonnegative quantity, we must have $D^G \geq 0$. If $\rho = \nu$, then the constant curve $\gamma(t) = \rho$ is a $G$-dual geodesic, since this curve solves $\dot{\gamma}(t) = \varphi^G_{\gamma(t)}([0])$. Hence, $D^G(\rho \mid \rho) = 0$.

If $D^G(\rho \mid \nu) = 0$, then by assumption there is a $G$-dual geodesic $\gamma$ connecting $\rho$ to $\nu$ such that $\int_0^1 t g^G_{\gamma(t)}(\dot{\gamma}(t), \dot{\gamma}(t)) \, \mathrm{d}t = 0$. Hence, $g^G_{\gamma(t)}(\dot{\gamma}(t), \dot{\gamma}(t)) = 0$ for almost all $t \in [0, 1]$, so $\dot{\gamma}(t) = 0$ for almost all $t$. Hence, $\gamma$ is constant, implying $\rho = \nu$.

If there is no $G$-dual geodesic connecting $\mu$ and $\nu$, then $D^G(\mu \mid \nu) = \infty$.

The asymmetric and lack of triangle inequality are well known for the KL divergence, i.e., $G = \mathrm{FR}$.

2. For two Gaussians $\mu \sim \mathcal{N}(m, \Sigma)$ and $\mu_1 \sim \mathcal{N}(m_1, \Sigma_1)$, we have $\mu \otimes \mu_1 \sim \mathcal{N}((m, m_1), \mathrm{diag}(\Sigma, \Sigma_1))$ and analogously for $\nu \otimes \nu_1$. To prove this statement, we only have to prove that the terms appearing in this formula act blockwise on block-diagonal matrices. This is, of course, true for the trace operator, for the pseudo inverse $\dagger$, the squared Euclidean norm, and the quadratic forms involved. If the covariance matrices are simultaneously diagonalizable, as Theorem 4.11 requires, then this also holds for all the terms appearing in the formula for the KWKL divergence.

3. See (Datar & Ay, 2026b, Subsec. 3.4) for the WKL. Since the formula only differs by the positive factor $\kappa_g$ at certain places, the continuity arguments are identical.

4. This statement is a sort of "data-processing equality", showing that $D^G$ inherits invariance properties from the underlying Onsager operator $\varphi^G$, e.g., that the Wasserstein-KL divergence is invariant under rigid transformations applied to both input measures simultaneously. In contrast, the Chentsov uniqueness theorem makes it very unlikely that state-space-aware KL divergences satisfy a data-processing inequality like the KL.

**Lemma A.1.** *If the Onsager operator is equivariant under the action of a Riemannian isometry $\Phi \in \mathrm{Isom}(M, g)$, in the sense that*

$$\Phi_\# \varphi^G_\mu([f]) = \varphi^G_{\Phi_\#\mu}([f \circ \Phi^{-1}]) \qquad \forall \mu \in \mathcal{P}^\infty_+(M), \forall [f] \in \mathcal{C}^\infty(M)/\mathbb{R}, \tag{17}$$

*then $D^G(\Phi_\#\mu \mid \Phi_\#\nu) = D^G(\mu \mid \nu)$ for all $\mu, \nu \in \mathcal{P}^\infty_+(M)$.*

*Proof.* For $\mu \in \mathcal{P}^\infty_+(M)$ we write $\mu = \rho\mu_g$. Since $\Phi \in \mathrm{Isom}(M, g)$, we have $\Phi_\#\mu_g = \mu_g$ and thus

$$\Phi_\#\mu = (\rho \circ \Phi^{-1})\mu_g.$$

In particular, $\Phi_\#(\mathcal{S}^\infty_0(M)) \subset \mathcal{S}^\infty_0(M)$, since

$$\left(\Phi_\#(f\mu_g)\right)(M) = \int_M (f \circ \Phi^{-1}) \, \mathrm{d}\mu_g = \int_M f \, \mathrm{d}\mu_g = 0$$

for all $f \in \mathcal{C}^\infty(M)$.

We divide the remainder of the proof into four steps.

(a) *Equivariance of $\Psi$.* For $f \in \mathcal{C}^\infty(M)$ we have

$$\mathbb{E}_{\Phi_\#\mu}[f \circ \Phi^{-1}] = \int_M f \circ \Phi^{-1} \, \mathrm{d}(\Phi_\#\mu) = \int_M f \, \mathrm{d}\mu = \mathbb{E}_\mu[f],$$

so that

$$\Psi_{\Phi_\#\mu}([f \circ \Phi^{-1}]) = f \circ \Phi^{-1} - \mathbb{E}_{\Phi_\#\mu}[f \circ \Phi^{-1}] = (f - \mathbb{E}_\mu[f]) \circ \Phi^{-1} = \Psi_\mu([f]) \circ \Phi^{-1}.$$

(b) *Invariance of $g^G$.* Let $\mu \in \mathcal{P}_+^\infty(M)$ and $a = \varphi_\mu^G([f]) \in \mathcal{S}_0^\infty(M)$ and $b = \varphi_\mu^G([g]) \in \mathcal{S}_0^\infty(M)$ for $[f], [g] \in \mathcal{C}^\infty(M)/\mathbb{R}$. Then,

$$g_{\Phi_\#\mu}(\Phi_\#a, \Phi_\#b) = \left\langle \Phi_\#a, \Psi_{\Phi_\#\mu}\left((\varphi_{\Phi_\#\mu}^G)^{-1}(\Phi_\#b)\right)\right\rangle_{L^2(M,\mu_g)}.$$

By (17),

$$(\varphi_{\Phi_\#\mu}^G)^{-1}(\Phi_\#b) = (\varphi_{\Phi_\#\mu}^G)^{-1}(\Phi_\#\varphi_\mu^G([g])) = [g \circ \Phi^{-1}],$$

so, using the equivariance of $\Psi$, that the above term simplifies to

$$\int_M \left(\varphi_\mu^G([f]) \circ \Phi^{-1}\right) \cdot \left(\Psi_\mu([g]) \circ \Phi^{-1}\right) \mathrm{d}\mu_g = \int_M \varphi_\mu^G([f]) \cdot \Psi_\mu([g]) \, \mathrm{d}\mu_g = g_\mu(a, b).$$

This shows that $\Phi_\#$ is an isometry of $(P_+^\infty, g^G)$.

(c) *Invariance of dual geodesics* Let $\gamma$ be a dual geodesic from $\mu$ to $\nu$, set $a := \varphi_\mu^G([f])$, and let $\tilde{\gamma} := \Phi_\#\gamma$. Then, for $t \in \mathbb{R}$, we have

$$\dot{\tilde{\gamma}}(t) = \Phi_\#\dot{\gamma}(t) = \Phi_\#\varphi_{\gamma(t)}^G\left((\varphi_\mu^G)^{-1}(a))\right) \overset{(17)}{=} \varphi_{\Phi_\#\gamma(t)}^G([f \circ \Phi^{-1}])$$

$$\overset{(17)}{=} \varphi_{\tilde{\gamma}(t)}^G\left(\left(\varphi_{\Phi_\#\mu}^G\right)^{-1}(\Phi_\#\varphi_\mu^G[f])\right) = \varphi_{\tilde{\gamma}(t)}^G\left(\left(\varphi_{\Phi_\#\mu}^G\right)^{-1}(\Phi_\#a)\right),$$

so $\tilde{\gamma}$ is the dual $G$-geodesic starting at $\Phi_\#\mu$ with initial velocity $\Phi_\#a$ and taking the value $\Phi_\#\nu$ at $t = 1$.

(d) *Equality of divergences.* Take $\mu, \nu \in \mathcal{P}_+^\infty(M)$. For any $G$-dual geodesic from $\mu$ to $\nu$, let $\tilde{\gamma} := \Phi_\#\gamma$. Then, for all $t \in [0, 1]$,

$$g_{\tilde{\gamma}(t)}^G\left(\dot{\tilde{\gamma}}(t), \dot{\tilde{\gamma}}(t)\right) = g_{\Phi_\#\gamma(t)}\left(\Phi_\#\dot{\gamma}(t), \Phi_\#\dot{\gamma}(t)\right) = g_{\gamma(t)}(\dot{\gamma}(t), \dot{\gamma}(t)).$$

and so taking the infimum over all such $G$-dual geodesics yields

$$D^G(\Phi_\#\mu \mid \Phi_\#\nu) \leq D^G(\mu \mid \nu).$$

Replacing $\Phi$ by $\Phi^{-1}$ in the preceding discussion yields the reverse inequality and thus the desired statement.

$\square$

*Example* A.2 (Invariant subgroups of isometries). We have $\mathrm{Isom}(\mathcal{S}^{d-1}) = O(d)$. For the formal extension to $M = \mathbb{R}^d$, we have $\mathrm{Isom}(\mathbb{R}^d) = \mathbb{R}^d \rtimes O(d)$ (the Euclidean group).

(a) *Fisher-Rao metric.* We have seen above that for $G = \mathrm{FR}$, (17) holds for any isometry $\Phi$.

(b) *Otto metric.* For any $g \in \mathcal{C}^\infty(M)$ we have

$$\int_M g \, \mathrm{d}\left[\Phi_\#\varphi_\mu^O([f])\right] = -\int_M g \circ \Phi \, \mathrm{div}_\mu(\mathrm{grad}\, f) \, \mathrm{d}\mu = \int_M \langle \mathrm{grad}(g \circ \Phi), \mathrm{grad}(f)\rangle \, \mathrm{d}\mu$$

$$= \int_M \langle \mathrm{grad}(g) \circ \Phi, \mathrm{grad}(f \circ \Phi^{-1}) \circ \Phi\rangle \, \mathrm{d}\mu$$

$$= \int_M \langle \mathrm{grad}(g), \mathrm{grad}(f \circ \Phi^{-1})\rangle \, \mathrm{d}\Phi_\#\mu$$

$$= -\int_M g \operatorname{div}_{\Phi_\# \mu}(\operatorname{grad}(f \circ \Phi^{-1})) \, \mathrm{d}\Phi_\# \mu$$

$$= \int_M g \, \mathrm{d}\left[\varphi^O_{\Phi_\# \mu}([f \circ \Phi^{-1}])\right],$$

so, again (17) holds for all isometries $\Phi \in \mathrm{Isom}(M, g)$.

(c) If the mobility function is indeed a scalar, then the same reasoning applies to the Otto metric with non-linear mobility.

(d) For the (normalized) Stein geometry on $M = \mathbb{R}^d$, we can only take $\Phi \in \mathrm{Isom}(\mathbb{R}^d)$ that preserve the kernel, that is, those with $K(\Phi x, \Phi y) = K(x, y)$ for all $x, y \in \mathbb{R}^d$. If $K = \varphi(\|\cdot - \cdot\|_2^2)$ is radial, then this is fulfilled. For the generalized bilinear kernel, we have to restrict to $\Phi(x) = Ox + b$ with $O^\mathsf{T} AO = A$ (stabilizers of $A$).

(e) For Sinkhorn-type geometries built using the kernel $e^{-\frac{1}{\varepsilon} c(x,y)}$, we can only take isometries that preserve the cost $c$. If $c$ is a squared metric, then these are all the isometries. $\qquad \bullet$

$\qquad \square$

## A.3. Proof of the formula for the linearized Wasserstein KL divergence Theorem 4.1

*Proof.* The $\dot{H}^{-1}$ gradient flow of the potential energy $\mu \mapsto \mathbb{E}_\mu[f]$ reads $\dot{\rho}_t = -\Delta f$, so $\rho_t = \rho - t\Delta f$. If we let $t = 1$, then we obtain the inhomogeneous Poisson equation $-\Delta f = \nu - \rho$, whose solution is $f = \Phi * (\nu - \rho)$. Then, (6) becomes

$$
\begin{aligned}
D^{\dot{H}^{-1}}(\mu \mid \nu) &= \int_{\mathbb{R}^d} f(x)\nu(x) \, \mathrm{d}x - \int_0^1 \int_{\mathbb{R}^d} f(x) \left(\rho(x) - t\Delta f(x)\right) \mathrm{d}x \, \mathrm{d}t \\
&= \int_{\mathbb{R}^d} f(x) \big(\nu(x) - \rho(x)\big) \, \mathrm{d}x + \int_0^1 t \int_{\mathbb{R}^d} f(x)\Delta f(x) \, \mathrm{d}x \, \mathrm{d}t \\
&= \frac{1}{2} \int_{\mathbb{R}^d} f(x) \big(\nu(x) - \rho(x)\big) \, \mathrm{d}x \\
&= \frac{1}{2} \int_{\mathbb{R}^d} \int_{\mathbb{R}^d} \Phi(x - y) \, \mathrm{d}(\mu - \nu)(x) \, \mathrm{d}(\mu - \nu)(y). \qquad \square
\end{aligned}
$$

## A.4. Proof of Lemma 4.8

*Proof.* 1. For the probability measures $(\mu_t)_{t \geq 0}$ associated to the densities $(\rho_t)_{t \geq 0}$, we have by (Ambrosio et al., 2008, Lemma 8.1.4) that $\mu_t = (X_t)_\# \mu_0$, where the flow $(X_t \colon \mathbb{R}^d \to \mathbb{R}^d)_{t \geq 0}$ solves

$$
\begin{cases}
\dot{X}_t = v_t \circ X_t, \\
X_0 = \mathrm{id} \,.
\end{cases}
$$

Then, there exist matrices $A_t \in \mathbb{R}^{d \times d}$ and vectors $a_t \in \mathbb{R}^d$ such that $X_t(x) = A_t x + a_t$ (where $\dot{A}_t = S_t A_t$, $A_0 = I$ and $\dot{a}_t = S_t a_t + s_t$, $a_0 = 0$). Now, suppose that $\rho_0 \sim E(m_0, \Sigma_0, g)$. Then, Remark 4.4 implies for any $t > 0$ there exist $m_t \in \mathbb{R}^d$ and $\Sigma_t \in \mathrm{Sym}_+(\mathbb{R}; d)$ such that $\rho_t \sim E(m_t, \Sigma_t, g)$.

2. We now derive the ODE (10) for the mean and covariance parameters $m_t$ and $\Sigma_t$. Using integration by parts, we have

$$
\dot{m}_t = \int_{\mathbb{R}^d} x\dot{\rho}_t(x) \, \mathrm{d}x = -\int_{\mathbb{R}^d} x\nabla \cdot \big(\rho_t(x)v_t(x)\big) \, \mathrm{d}x = \int_{\mathbb{R}^d} \rho_t(x)(S_t x + s_t) \, \mathrm{d}x = S_t m_t + s_t,
$$

and, similarly,

$$
\begin{aligned}
\dot{\Sigma}_t &= \partial_t \int_{\mathbb{R}^d} (x - m_t)(x - m_t)^\mathsf{T} \rho_t(x) \, \mathrm{d}x \\
&= \int_{\mathbb{R}^d} (x - m_t)(x - m_t)^\mathsf{T} \partial_t \rho_t(x) \, \mathrm{d}x - 2\dot{m}_t \underbrace{\int_{\mathbb{R}^d} (x - m_t)^\mathsf{T} \rho_t(x) \, \mathrm{d}x}_{=0}
\end{aligned}
$$

$$= -\int_{\mathbb{R}^d} (x - m_t)(x - m_t)^\mathsf{T} \nabla \cdot \big(v_t(x)\rho_t(x)\big)\,\mathrm{d}x$$

$$= \int_{\mathbb{R}^d} \big((x - m_t)v_t(x)^\mathsf{T} + v_t(x)(x - m_t)^\mathsf{T}\big)\rho_t(x)\,\mathrm{d}x$$

$$= \int_{\mathbb{R}^d} \big((x - m_t)(S_t x + s_t)^\mathsf{T} + (S_t x + s_t)(x - m_t)^\mathsf{T}\big)\rho_t(x)\,\mathrm{d}x$$

$$= \int_{\mathbb{R}^d} \big((x - m_t)x^\mathsf{T}S_t^\mathsf{T} + S_t x(x - m_t)^\mathsf{T}\big)\rho_t(x)\,\mathrm{d}x$$

$$= \int_{\mathbb{R}^d} \big((x - m_t)(x - m_t)^\mathsf{T}S_t^\mathsf{T} + S_t(x - m_t)(x - m_t)^\mathsf{T}\big)\rho_t(x)\,\mathrm{d}x$$

$$= \Sigma_t S_t^\mathsf{T} + S_t \Sigma_t. \qquad \square$$

### A.5. Proof of Theorem 4.10

*Proof.* Since $v^{\mathrm{O}}(\mu, \nabla f) = \nabla f$, Lemma 4.8 gives us the linear time-invariant and decoupled system

$$\begin{cases} \dot{m}_t = Bm_t + b, \\ \dot{\Sigma}_t = 2\,\mathrm{Sym}(B\Sigma_t). \end{cases} \tag{18}$$

which can be analytically solved to obtain

$$m_1 = e^B m_0 + \big(e^B B^\dagger + B^\perp - B^\dagger\big)\,b,$$

$$\Sigma_1 = e^B \Sigma_0 e^B.$$

Under the constraints that $\Sigma_0 \succ 0$ and $\Sigma_1 \succ 0$, these equations possess a unique solution (Kučera, 1973, Thm. 5) given by

$$B = \log\left(\Sigma_0^{-\frac{1}{2}}\left(\Sigma_0^{\frac{1}{2}}\Sigma_1\Sigma_0^{\frac{1}{2}}\right)^{\frac{1}{2}}\Sigma_0^{-\frac{1}{2}}\right), \tag{19}$$

$$b = \big((e^B - I)B^\dagger + B^\perp\big)^{-1}(m_1 - e^B m_0). \tag{20}$$

gives us the final result (12). $\qquad \square$

### A.6. Proof of the Kalman-Wasserstein-KL formula in Theorem 4.11

Now, we prove Theorem 4.11.

*Proof.* 1. First, we solve the mean-covariance ODE. By Equations (9) and (10) the geodesic $\gamma(t) \sim E(m_t, \Sigma_t, g)$ follows

$$\begin{cases} \dot{m}_t = (\kappa_g \Sigma_t + \lambda I)(Bm_t + b), \\ \dot{\Sigma}_t = 2\,\mathrm{Sym}\big((\kappa_g \Sigma_t + \lambda I)B\Sigma_t\big). \end{cases} \tag{21}$$

We choose $B$ such that $B$ and $\Sigma_0$ are simultaneously diagonalizable with

$$\Sigma_0 = Q\,\mathrm{diag}(s_0)Q^{-1} \qquad \text{and} \qquad B = Q\,\mathrm{diag}(\beta_i)Q^{-1}$$

From now on, we will only work in the $Q$-basis: we can write $\Sigma_t = Q\,\mathrm{diag}(s_t)Q^{-1}$, since if $\Sigma_t$ is the unique solution, then $\partial_t Q^{-1}\Sigma_t Q = 2\,\mathrm{Sym}((\kappa_g Q^{-1}\Sigma_t Q + \lambda I)\,\mathrm{diag}(\beta)Q^{-1}\Sigma_t Q)$ is diagonal. Further, let $\tilde{b} := Q^{-1}b$, and $\tilde{m}_t := Q^{-1}m_t$, we obtain

$$\begin{cases} (\dot{\tilde{m}}_t)_i = (\kappa_g(s_t)_i + \lambda)(\beta_i(\tilde{m}_t)_i + \tilde{b}_i), \\ (\dot{s}_t)_i = 2(\kappa_g(s_t)_i + \lambda)(s_t)_i \odot \beta_i, \end{cases}$$

where $\odot$ denotes elementwise multiplication of vectors. This system can be solved componentwise: the covariance equation has the closed-form solution

$$(s_i)_t = \frac{\lambda(s_0)_i}{(\kappa_g(s_0)_i + \lambda)e^{-2\lambda\beta_i t} - \kappa_g(s_0)_i}, \qquad t < \begin{cases} \frac{1}{2\lambda\beta_i}\ln\left(1 + \frac{\lambda}{\kappa_g(s_0)_i}\right), & \beta_i > 0, \\ \infty, & \beta_i \le 0. \end{cases}$$

(Equivalently, $\Sigma_t = \lambda\Sigma_0\big((\lambda I + \kappa_g\Sigma_0)e^{-2\lambda Bt} - \kappa_g\Sigma_0\big)^{-1}$.) We will see that the upper bound on the existence interval is larger than 1, so that everything is well-defined.

Plugging this solution into the equation for the mean yields, after some algebra

$$
(\tilde m_t)_i = \begin{cases} (\tilde m_0)_i + t\big(\kappa_g(s_0)_i + \lambda\big)\tilde b_i, & \text{if } \beta_i = 0, \\ \big((\tilde m_0)_i + \frac{\tilde b_i}{\beta_i}\big)\sqrt{\frac{\lambda}{(\kappa_g(s_0)_i + \lambda)e^{-2\lambda\beta_i t} - \kappa_g(s_0)_i}} - \frac{\tilde b_i}{\beta_i}, & \text{else.} \end{cases}
\tag{22}
$$

2. Now, we solve for $B$ and $b$.

Solving for $B$ yields

$$
\Sigma_1 = \lambda\Sigma_0\left((\lambda I + \kappa_g\Sigma_0)e^{-2\lambda B} - \kappa_g\Sigma_0\right)^{-1}
$$
$$
\iff e^{2\lambda B} = \left(\lambda\Sigma_0^{-1} + \kappa_g I\right)\left(\lambda\Sigma_1^{-1} + \kappa_g I\right)^{-1}.
$$

Since $\Sigma_0$ and $\Sigma_1$ are symmetric positive definite matrices and $\lambda, \kappa_g > 0$, the principal matrix logarithm of the right-hand side exists and is unique, and we obtain

$$
B = \frac{1}{2\lambda}\ln\left(\left(\lambda\Sigma_0^{-1} + \kappa_g I\right)\left(\lambda\Sigma_1^{-1} + \kappa_g I\right)^{-1}\right).
$$

The matrix $B$ is symmetric since $\Sigma_0\Sigma_1 = \Sigma_1\Sigma_0$. Hence,

$$
\beta_i = \frac{1}{2\lambda}\ln\left(\frac{\frac{\lambda}{(s_0)_i} + \kappa_g}{\frac{\lambda}{(s_1)_i} + \kappa_g}\right), \qquad i \in \{1, \ldots, d\}.
\tag{23}
$$

and thus $\beta_i = 0$ if and only if $(s_0)_i = (s_1)_i$ for $i \in \{1, \ldots, d\}$. Hence,

$$
(\Sigma_1 - \Sigma_0)^\perp = I - (\Sigma_1 - \Sigma_0)(\Sigma_1 - \Sigma_0)^\dagger
$$
$$
= I - Q\operatorname{diag}(s_1 - s_0)Q^{-1}Q\operatorname{diag}\left(\frac{1}{s_1 - s_0}\mathbb{1}_{s_1 \neq s_0}\right)Q^{-1}
$$
$$
= I - Q\operatorname{diag}(1_{(s_1)_i \neq (s_0)_i})Q^{-1} = Q\operatorname{diag}(1_{(s_1)_i = (s_0)_i})Q^{-1}.
$$

Furthermore, we can solve for $\tilde b$ in terms of $\mu$ and $\nu$:

$$
\tilde b_i = \begin{cases} \frac{(\tilde m_1)_i - (\tilde m_0)_i}{\kappa_g(s_0)_i + \lambda}, & \text{if } \beta_i = 0, \\ \beta_i\frac{(\tilde m_1)_i\sqrt{(\xi_1)_i} - (\tilde m_0)_i\sqrt{\lambda}}{\sqrt{\lambda} - \sqrt{(\xi_1)_i}}, & \text{else.} \end{cases}
\tag{24}
$$

where we used the shorthand $(\xi_t)_i := (\kappa_g(s_0)_i + \lambda)e^{-2\lambda\beta_i t} - \kappa_g(s_0)_i$ for $i \in \{1, \ldots, d\}$. Furthermore, (23) implies

$$
e^{-2\lambda\beta_i t} = \left(\frac{\frac{\lambda}{(s_1)_i} + \kappa_g}{\frac{\lambda}{(s_0)_i} + \kappa_g}\right)^t, \qquad i \in \{1, \ldots, d\},
\tag{25}
$$

so $(\xi_1)_i = \lambda\frac{(s_0)_i}{(s_1)_i}$ for all $i \in \{1, \ldots, d\}$.

3. Now, we can evaluate the divergence formula from (11):

$$
D^{\mathrm{KW}}(\mu \mid \nu) = \frac{\kappa_g}{2}\operatorname{tr}(B\Sigma_1) + \frac{1}{2}m_1^\mathsf{T}(Bm_1 + 2b) - \int_0^1 \frac{\kappa_g}{2}\operatorname{tr}(B\Sigma_t) + \frac{1}{2}m_t^\mathsf{T}(Bm_t + 2b)\,\mathrm{d}t.
$$

Splitting the sum into the $\beta_i = 0$ and $\beta_i \neq 0$ contributions and plugging in (22) yields

$$
\frac{\kappa_g}{2}\operatorname{tr}(B\Sigma_t) + \frac{1}{2}m_t^\mathsf{T}(Bm_t + 2b) = \frac{1}{2}\sum_{i=1}^d \kappa_g\beta_i(s_t)_i + (\tilde m_t)_i(\beta_i(\tilde m_t)_i + 2\tilde b_i)
$$

$$= \sum_{\substack{i=1 \\ \beta_i=0}}^{d} \left((\tilde{m}_0)_i + t\big((\kappa_g s_0)_i + \lambda\big)\tilde{b}_i\right)\tilde{b}_i$$

$$+ \frac{1}{2} \sum_{\substack{i=1 \\ \beta_i\neq0}}^{d} \beta_i \underbrace{\left(\kappa_g (s_t)_i + (\tilde{m}_t)_i \left((\tilde{m}_t)_i + 2\frac{(\tilde{m}_1)_i\sqrt{(\xi_1)_i} - (\tilde{m}_0)_i\sqrt{\lambda}}{\sqrt{\lambda} - \sqrt{(\xi_1)_i}}\right)\right)}_{=:(\star)}.$$

We have

$$(\star) = \kappa_g \frac{\lambda(s_0)_i}{\xi_t} + \left(\left((\tilde{m}_0)_i + \frac{(\tilde{m}_1)_i\sqrt{\xi_1} - (\tilde{m}_0)_i\sqrt{\lambda}}{\sqrt{\lambda} - \sqrt{\xi_1}}\right)\frac{\sqrt{\lambda}}{\sqrt{\xi_t}} - \frac{(\tilde{m}_1)_i\sqrt{\xi_1} - (\tilde{m}_0)_i\sqrt{\lambda}}{\sqrt{\lambda} - \sqrt{\xi_1}}\right)$$

$$\cdot \left(\left((\tilde{m}_0)_i + \frac{(\tilde{m}_1)_i\sqrt{\xi_1} - (\tilde{m}_0)_i\sqrt{\lambda}}{\sqrt{\lambda} - \sqrt{\xi_1}}\right)\frac{\sqrt{\lambda}}{\sqrt{\xi_t}} + \frac{(\tilde{m}_1)_i\sqrt{\xi_1} - (\tilde{m}_0)_i\sqrt{\lambda}}{\sqrt{\lambda} - \sqrt{\xi_1}}\right)$$

$$= \kappa_g \frac{\lambda(s_0)_i}{\xi_t} + \left((\tilde{m}_0)_i + \frac{(\tilde{m}_1)_i\sqrt{\xi_1} - (\tilde{m}_0)_i\sqrt{\lambda}}{\sqrt{\lambda} - \sqrt{\xi_1}}\right)^2 \frac{\lambda}{\xi_t} - \left(\frac{(\tilde{m}_1)_i\sqrt{\xi_1} - (\tilde{m}_0)_i\sqrt{\lambda}}{\sqrt{\lambda} - \sqrt{\xi_1}}\right)^2.$$

The $\beta_i = 0$ part of $D^{\mathrm{KW}}(\mu \mid \nu)$ is

$$\sum_{\substack{i=1 \\ \beta_i=0}}^{d} \frac{(\tilde{m}_1)_i((\tilde{m}_1)_i - (\tilde{m}_0)_i)}{\kappa_g(s_0)_i + \lambda} - \int_0^1 ((\tilde{m}_0)_i + t((\tilde{m}_1)_i - (\tilde{m}_0)_i))\,\mathrm{d}t \cdot \frac{(\tilde{m}_1)_i - (\tilde{m}_0)_i}{\kappa_g(s_0)_i + \lambda}$$

$$= \sum_{\substack{i=1 \\ \beta_i=0}}^{d} \left((\tilde{m}_1)_i - \frac{(\tilde{m}_1)_i + (\tilde{m}_0)_i}{2}\right)\frac{(\tilde{m}_1)_i - (\tilde{m}_0)_i}{\kappa_g(s_0)_i + \lambda} = \frac{1}{2}\sum_{\substack{i=1 \\ (s_0)_i=(s_1)_i}}^{d} \frac{((\tilde{m}_1)_i - (\tilde{m}_0)_i)^2}{\kappa_g(s_0)_i + \lambda}$$

$$= \frac{1}{2}(m_1 - m_0)^{\mathsf{T}}(\Sigma_1 - \Sigma_0)^{\perp}(\kappa_g\Sigma_0 + \lambda I)^{-1}(\Sigma_1 - \Sigma_0)^{\perp}(m_1 - m_0).$$

For $i$-th summand of the $\beta_i \neq 0$ part of $D^{\mathrm{KW}}(\mu \mid \nu)$, we first integrate the only $t$-dependent term:

$$\int_0^1 \frac{1}{(\xi_t)_i}\,\mathrm{d}t = \int_0^1 \frac{1}{(\kappa_g(s_0)_i + \lambda)e^{-2\lambda\beta_i t} - \kappa_g(s_0)_i}\,\mathrm{d}t$$

$$= \frac{1}{\kappa_g(s_0)_i 2\lambda\beta_i}\ln\left(\frac{\lambda}{(\xi_1)_i}\right) - \frac{1}{\kappa_g(s_0)_i}$$

$$= \frac{1}{2\lambda\beta_i\kappa_g(s_0)_i}\ln\left(\frac{(s_1)_i}{(s_0)_i}\right) - \frac{1}{\kappa_g(s_0)_i}$$

Thus,

$$\frac{1}{(\xi_1)_i} - \int_0^1 \frac{1}{(\xi_t)_i}\,\mathrm{d}t = \frac{\kappa_g(s_1)_i + \lambda}{\lambda\kappa_g(s_0)_i} - \frac{1}{2\lambda\beta_i\kappa_g(s_0)_i}\ln\left(\frac{(s_1)_i}{(s_0)_i}\right).$$

Hence, the complete contribution of each $\beta_i \neq 0$ term is (note that the affine term cancels)

$$\frac{\lambda}{2}\beta_i\left(\kappa_g(s_0)_i + \left((\tilde{m}_0)_i + \frac{(\tilde{m}_1)_i\sqrt{(\xi_1)_i} - (\tilde{m}_0)_i\sqrt{\lambda}}{\sqrt{\lambda} - \sqrt{(\xi_1)_i}}\right)^2\right)\left(\frac{1}{(\xi_1)_i} - \int_0^1 \frac{1}{(\xi_t)_i}\,\mathrm{d}t\right).$$

We have

$$(\tilde{m}_0)_i + \frac{(\tilde{m}_1)_i\sqrt{(\xi_1)_i} - (\tilde{m}_0)_i\sqrt{\lambda}}{\sqrt{\lambda} - \sqrt{(\xi_1)_i}} = \frac{\sqrt{s_0}}{\sqrt{s_1} - \sqrt{s_0}}((\tilde{m}_1)_i - (\tilde{m}_0)_i)$$

and thus the full contribution reads

$$\sum_{\substack{i=1 \\ (s_0)_i\neq(s_1)_i}}^{d} \left(\kappa_g(s_0)_i + \frac{(s_0)_i((\tilde{m}_1)_i - (\tilde{m}_0)_i)^2}{\left(\sqrt{s_1} - \sqrt{s_0}\right)^2}\right)\left(\frac{1}{4\lambda}\ln\left(\frac{\frac{\lambda}{(s_0)_i} + \kappa_g}{\frac{\lambda}{(s_1)_i} + \kappa_g}\right)\frac{\kappa_g(s_1)_i + \lambda}{(s_0)_i\kappa_g} - \frac{1}{4\kappa_g(s_0)_i}\ln\left(\frac{(s_1)_i}{(s_0)_i}\right)\right)$$

$$= \frac{1}{4\lambda} \sum_{\substack{i=1 \\ (s_0)_i \neq (s_1)_i}}^{d} \left(1 + \frac{((\tilde{m}_1)_i - (\tilde{m}_0)_i)^2}{\kappa_g \left(\sqrt{s_1} - \sqrt{s_0}\right)^2}\right) \left(\kappa_g(s_1)_i \ln \left(\frac{(s_1)_i}{(s_0)_i}\right) + (\kappa_g(s_1)_i + \lambda) \ln \left(\frac{\lambda + \kappa_g(s_0)_i}{\lambda + \kappa_g(s_1)_i}\right)\right).$$

In matrix form, we obtain

$$\frac{1}{4\lambda} \operatorname{tr}\left((\kappa_g \Sigma_1 + \lambda I) \ln \left((\kappa_g \Sigma_0 + \lambda I)(\kappa_g \Sigma_1 + \lambda I)^{-1}\right)\right) + \frac{\kappa_g}{4\lambda} \operatorname{tr}\left(\Sigma_1 \ln \left(\Sigma_1 \Sigma_0^{-1}\right)\right)$$

$$+ \frac{1}{4\lambda}(m_1 - m_0)^\mathsf{T} P \left(\Sigma_1 \ln \left(\Sigma_1 \Sigma_0^{-1}\right) + \frac{1}{\kappa_g}(\kappa_g \Sigma_1 + \lambda I) \ln \left((\lambda I + \kappa_g \Sigma_0)(\lambda I + \kappa_g \Sigma_1)^{-1}\right)\right)$$

$$\cdot P \left(\left(\sqrt{\Sigma_1} - \sqrt{\Sigma_0}\right)^\dagger\right)^2 (m_1 - m_0),$$

where $P := (\Sigma_1 - \Sigma_0)(\Sigma_1 - \Sigma_0)^\dagger$. $\qquad\square$

We now show that one can remove the assumption of simultaneous commutativity in Theorem 4.11 for Gaussian distributions.

**Theorem A.3** (KWKL for Gaussians with arbitrary covariance matrices)**.** *Let $\mu \sim \mathcal{N}(m_0, \Sigma_0)$ and $\nu \sim \mathcal{N}(m_1, \Sigma_1)$. With $\Delta m := m_1 - m_0$ we have*

$$D^{\mathrm{KW}}(\mu \mid \nu) = \frac{1}{2} \operatorname{tr}(B\Sigma_1) - \frac{1}{4} \log \left(\det \left((\Sigma_1 + \lambda I)(\Sigma_0 + \lambda I)^{-1}\right)\right) + \Delta m^\mathsf{T} W^{-\mathsf{T}} K W^{-1} \Delta m,$$

*where $B$ is defined in (27), and $\Sigma_t^{-1} = e^{-\lambda t B}\left(\Sigma_0^{-1} + \lambda^{-1}I\right)e^{-\lambda t B} - \lambda^{-1}I$, and $W := \int_0^1 (\Sigma_t + \lambda I)\Psi(t, 0)\,\mathrm{d}t$ is assumed to be invertible, $K := \int_0^1 t\Psi(t, 0)^\top (\Sigma_t + \lambda I)\Psi(t, 0)\,\mathrm{d}t$, and $\Psi$ is the fundamental matrix defined by $\partial_t \Psi(t, s) = B(\Sigma_t + \lambda I)\Psi(t, s)$ and $\Psi(s, s) = I$.*

*Proof.* Again, we consider the quadratic potential $f(x) = \frac{1}{2}x^\top B x + b^\top x + c$, where $B$ is symmetric. The mean and covariance ODE is

$$\begin{cases} \dot{m}_t = (\Sigma_t + \lambda I)(Bm_t + b), \\ \dot{\Sigma}_t = (\Sigma_t + \lambda I)B\Sigma_t + \Sigma_t B(\Sigma_t + \lambda I). \end{cases} \tag{26}$$

We can solve the covariance equation without diagonalization. In fact, using the substitution $M_{t,\lambda} := \Sigma_t^{-1} + \frac{1}{\lambda}I$, we have $\dot{M}_{t,\lambda} = -2\lambda \operatorname{Sym}(M_{t,\lambda}B)$. Hence, $M_{t,\lambda} = e^{-\lambda t B}M_{0,\lambda}e^{-\lambda t B}$ and so

$$\Sigma_t^{-1} = e^{-\lambda t B}\left(\Sigma_0^{-1} + \lambda^{-1}I\right)e^{-\lambda t B} - \lambda^{-1}I.$$

Determining $B \in \operatorname{Sym}(\mathbb{R}; d)$ from the endpoints $\Sigma_0$ and $\Sigma_1$ yields

$$B = -\frac{1}{\lambda} \log \left(M_{0,\lambda}^{-\frac{1}{2}} \left(M_{0,\lambda}^{\frac{1}{2}} M_{1,\lambda} M_{0,\lambda}^{\frac{1}{2}}\right)^{\frac{1}{2}} M_{0,\lambda}^{-\frac{1}{2}}\right). \tag{27}$$

Now, we first consider the centered case where $m_0 = m_1 = 0$. By (11),

$$D^{\mathrm{KW}}(\mu \mid \nu) = \frac{1}{2} \operatorname{tr}(B\Sigma_1) - \frac{1}{2} \int_0^1 \operatorname{tr}(B\Sigma_t)\,\mathrm{d}t.$$

By well-known identities from matrix calculus, we have

$$\frac{\mathrm{d}}{\mathrm{d}t} \log(\det(\Sigma_t + \lambda I)) = \operatorname{tr}\left((\Sigma_t + \lambda I)^{-1}\dot{\Sigma}_t\right) \overset{(26)}{=} \operatorname{tr}\left(B\Sigma_t + (\Sigma_t + \lambda I)^{-1}\Sigma_t B(\Sigma_t + \lambda I)\right) = 2\operatorname{tr}(B\Sigma_t).$$

Hence,

$$D^{\mathrm{KW}}(\mu \mid \nu) = \frac{1}{2} \operatorname{tr}(B\Sigma_1) - \frac{1}{4} \log \left(\det \left((\Sigma_1 + \lambda I)(\Sigma_0 + \lambda I)^{-1}\right)\right).$$

Now, we turn to the general case. Changing coordinates, we define $r_t := Bm_t + b$. The mean equation in (26) becomes $\dot{r}_t = B(\Sigma_t + \lambda I)r_t$. Letting $\Psi$ be the corresponding fundamental matrix, we have $r_t = \Psi(t, 0)r_0$. Thus, $\Delta m = \int_0^1 (\Sigma_t + \lambda I)r_t\,\mathrm{d}t = \int_0^1 (\Sigma_t + \lambda I)\Psi(t, 0)\,\mathrm{d}t\,r_0 = Wr_0$. Thus $r_0 = W^{-1}\Delta m$ and $b = r_0 - Bm_0$. Thus, the missing additive contribution of the mean from (11) is

$$\int_0^1 t \cdot r_t^\top (\Sigma_t + \lambda I)r_t\,\mathrm{d}t = \Delta m^\top W^{-\top} \int_0^1 t\Psi(t, 0)^\top (\Sigma_t + \lambda I)\Psi(t, 0)\,\mathrm{d}t \cdot W^{-1}\Delta m. \qquad\square$$

## B. Elementary calculations with elliptic distributions

In this section, we perform some elementary computations relating to elliptic distributions that are used in the main text.

**Lemma B.1** (Covariance of elliptic distributions). *Let $X \sim \mu$ be an $E(m, \Sigma, g)$-distributed random variable. Then,* $\mathbb{E}[X] = m$ *and* $\mathbb{V}[X] = \frac{1}{d} \frac{\pi^{\frac{d}{2}}}{\Gamma(\frac{d}{2})} \int_0^\infty r^{\frac{d}{2}} g(r) \, \mathrm{d}r \cdot \Sigma$.

*Proof.* Let $Y := \Sigma^{-\frac{1}{2}}(X - m)$. Its law is $(\Sigma^{-\frac{1}{2}}(\cdot - m))_{\#}\mu$, whose density is $g \circ \|\cdot\|^2$. We have

$$\mathbb{E}[X] = \Sigma^{\frac{1}{2}} \underbrace{\mathbb{E}[Y]}_{=0} + m \int_{\mathbb{R}^d} g(\|y\|_2^2) \, \mathrm{d}y = m \frac{\pi^{\frac{d}{2}}}{\Gamma(\frac{d}{2})} \int_0^\infty r^{\frac{d}{2}-1} g(r) \, \mathrm{d}r = m,$$

where the first integral vanishes because the integrand is odd, as well as

$$\mathbb{V}[X] = \int_{\mathbb{R}^d} (x - m)(x - m)^\mathsf{T} p_{m, \Sigma, g}(x) \, \mathrm{d}x = \Sigma^{\frac{1}{2}} \left( \int_{\mathbb{R}^d} yy^\mathsf{T} g(\|y\|_2^2) \, \mathrm{d}y \right) \Sigma^{\frac{1}{2}}.$$

Let $\Sigma_Y := \int_{\mathbb{R}^d} yy^\mathsf{T} g(\|y\|_2^2) \, \mathrm{d}y$. For any orthogonal matrix $O \in O(d)$, we have $O\Sigma_Y O^\mathsf{T} = \Sigma_Y$. Hence, there exists a $\kappa_g \in \mathbb{R}$ with $\Sigma_Y = \kappa_g \, \mathrm{id}_d$. We have $\kappa_g d = \mathrm{tr}(\Sigma_Y) = \int_{\mathbb{R}^d} \|y\|^2 g(\|y\|_2^2) \, \mathrm{d}y$, so $\mathbb{V}[X] = \kappa_g \Sigma$ with

$$\kappa_g = \frac{1}{d} \int_{\mathbb{R}^d} \|y\|^2 g(\|y\|^2) \, \mathrm{d}y = \frac{1}{d} \frac{\pi^{\frac{d}{2}}}{\Gamma(\frac{d}{2})} \int_0^\infty r^{\frac{d}{2}} g(r) \, \mathrm{d}r. \qquad \square$$

**Lemma B.2** (Potential energy of elliptic distribution). *If $\mu \sim E(m, \Sigma, g)$ and $f(x) = \frac{1}{2}x^\mathsf{T} Bx + b^\mathsf{T} x + c$, then*

$$\int_{\mathbb{R}^d} f \, \mathrm{d}\mu = \frac{\kappa_g}{2} \mathrm{tr}(B\Sigma) + \frac{1}{2} m^\mathsf{T} Bm + b^\mathsf{T} m + c.$$

*Proof.* Using the "whitening" substitution $y := \Sigma^{-\frac{1}{2}}(x - m)$ we have

$$\begin{aligned}
\int_{\mathbb{R}^d} f \, \mathrm{d}\mu &= \int_{\mathbb{R}^d} \left( \frac{1}{2}x^\mathsf{T} Bx + b^\mathsf{T} x + c \right) p_{m, \Sigma, g}(x) \, \mathrm{d}x \\
&= \frac{1}{2} \int_{\mathbb{R}^d} \left( m^\mathsf{T} + y^\mathsf{T} \Sigma^{\frac{1}{2}} \right) B \left( \Sigma^{\frac{1}{2}} y + m \right) g(\|y\|_2^2) \, \mathrm{d}y + b^\mathsf{T} m + c \\
&= \frac{1}{2} \mathrm{tr} \left( \Sigma^{\frac{1}{2}} B \Sigma^{\frac{1}{2}} \int_{\mathbb{R}^d} yy^\mathsf{T} g(\|y\|_2^2) \, \mathrm{d}y \right) + \frac{1}{2} m^\mathsf{T} Bm + b^\mathsf{T} m + c \\
&= \frac{\kappa_g}{2} \mathrm{tr}(B\Sigma) + \frac{1}{2} m^\mathsf{T} Bm + b^\mathsf{T} m + c,
\end{aligned}$$

using that $\int_{\mathbb{R}^d} yg(\|y\|_2^2) \, \mathrm{d}y = 0$, $\int_{\mathbb{R}^d} g(\|y\|_2^2) \, \mathrm{d}y = 1$ and the linearity and cyclic permutation property of the trace together with $y^\mathsf{T} Ay = \mathrm{tr}(Ayy^\mathsf{T})$ and Lemma B.1. $\qquad \square$

### B.1. Calculating the vector fields for elliptic distributions

We now verify the calculations from Example 4.7

**Lemma B.3** (Kalman-Wasserstein vector field for elliptic distribution). *The Kalman-Wasserstein vector field for an elliptic distribution is (9).*

$$v^{\mathrm{KW}}(p_{m, \Sigma, g}, \nabla f) = x \mapsto (\kappa_g \Sigma + \lambda \, \mathrm{id}_d)(Bx + b). \tag{28}$$

*Proof.* Comparing (8) with (3) yields the vector field

$$v^{\mathrm{KW}}(p_{m, \Sigma, g}, \nabla f) = \frac{m(p_{m, \Sigma, g})}{p_{m, \Sigma, g}} \nabla f = (\mathbb{V}[p_{m, \Sigma, g}] + \lambda I) \nabla f = (\kappa_g \Sigma + \lambda I_d) \nabla f. \qquad \square$$

### B.2. Fisher-Rao calculation

We show that

$$D^{\mathrm{FR}}(\mu \mid \nu) = \mathrm{KL}(\nu \mid \mu). \tag{29}$$

for $\mu, \nu \in \mathcal{P}_+^\infty(\mathbb{R}^n)$ such that

$$\int_{\mathbb{R}^n} \left| \ln\left(\frac{d\mu}{d\nu}\right) \right|^k \mathrm{d}(\mu + \nu) < \infty, \qquad k = 2. \tag{30}$$

Note that (30) holds for Gaussian distributions and for Student-t distributions $\mu, \nu$ with $v > 2$ degrees of freedom.

*Proof.* The FR geodesic is given by

$$\gamma(t) = \frac{\left(\frac{d\nu}{d\mu}\right)^t}{Z(t)} \mu = \rho_t \cdot \mu, \text{ where } Z(t) = \int_{\mathbb{R}^n} \left(\frac{d\nu}{d\mu}\right)^t \mathrm{d}\mu.$$

Note that if $\mu = f_\mu \mathcal{L}$ and $\nu = f_\nu \mathcal{L}$, then

$$Z(t) = \int_{\mathbb{R}^n} \left(\frac{d\nu}{d\mu}\right)^t \mathrm{d}\mu = \int_{\mathbb{R}^n} \left(\frac{f_\nu}{f_\mu}\right)^t f_\mu \,\mathrm{d}\mathcal{L} = \int_{\mathbb{R}^n} \left(\frac{f_\nu}{f_\mu}\right)^t f_\mu \,\mathrm{d}\mathcal{L} = \int_{\mathbb{R}^n} f_\nu^t f_\mu^{(1-t)} \,\mathrm{d}\mathcal{L}.$$

Note that $f_\nu^t f_\mu^{(1-t)}$ is differentiable with respect to $t$ and

$$\frac{\mathrm{d}}{\mathrm{d}t}\left(f_\nu^t f_\mu^{(1-t)}\right) = \left(f_\nu^t f_\mu^{(1-t)}\right) \ln\left(\frac{f_\nu}{f_\mu}\right).$$

Note that (30) implies the condition for $k = 1$ since $\mu + \nu$ has finite mass. The AM-GM inequality together with (30) for $k = 1$ justify differentiating under the integral sign, yielding

$$\frac{\mathrm{d}}{\mathrm{d}t} Z(t) = \int_{\mathbb{R}^n} \left(f_\nu^t f_\mu^{(1-t)}\right) \ln\left(\frac{f_\nu}{f_\mu}\right) \mathrm{d}\mathcal{L} = \int_{\mathbb{R}^n} \left(\frac{f_\nu}{f_\mu}\right)^t \ln\left(\frac{f_\nu}{f_\mu}\right) \mathrm{d}\mu = \int_{\mathbb{R}^n} \left(\frac{d\nu}{d\mu}\right)^t \ln\left(\frac{d\nu}{d\mu}\right) \mathrm{d}\mu.$$

This gives us

$$\frac{1}{Z(t)} \frac{\mathrm{d}}{\mathrm{d}t} Z(t) = \frac{1}{\int_{\mathbb{R}^n} \left(\frac{d\nu}{d\mu}\right)^t \mathrm{d}\mu} \int_{\mathbb{R}^n} \left(\frac{d\nu}{d\mu}\right)^t \ln\left(\frac{d\nu}{d\mu}\right) \mathrm{d}\mu = \int_{\mathbb{R}^n} \ln\left(\frac{d\nu}{d\mu}\right) \mathrm{d}\gamma(t).$$

Differentiating $\gamma$ with respect to time and using the above expression for $\frac{\dot{Z}(t)}{Z(t)}$, we get that

$$\dot{\gamma}(t) = \left(\frac{\left(\frac{d\nu}{d\mu}\right)^t \ln\left(\frac{d\nu}{d\mu}\right)}{Z(t)} - \frac{\left(\frac{d\nu}{d\mu}\right)^t \dot{Z}(t)}{Z(t)^2}\right) \mu = \left(\ln\left(\frac{d\nu}{d\mu}\right) - \frac{\dot{Z}(t)}{Z(t)}\right) \gamma(t)$$

$$= \left(\ln\left(\frac{d\nu}{d\mu}\right) - \int_{\mathbb{R}^n} \ln\left(\frac{d\nu}{d\mu}\right) \mathrm{d}\gamma(t)\right) \gamma(t)$$

$$= \left(\ln\left(\frac{d\nu}{d\mu}\right) - \mathbb{E}_{\gamma(t)}\left[\ln\left(\frac{d\nu}{d\mu}\right)\right]\right) \gamma(t).$$

Plugging this into the expression for $D^{\mathrm{FR}}$ yields

$$D^{\mathrm{FR}}(\mu \mid \nu) = \int_0^1 t \left(\int_{\mathbb{R}^n} \left(\ln\left(\frac{d\nu}{d\mu}\right) - \mathbb{E}_{\gamma(t)}\left[\ln\left(\frac{d\nu}{d\mu}\right)\right]\right)^2 \mathrm{d}\gamma(t)\right) \mathrm{d}t$$

$$= \int_0^1 t \cdot \mathbb{E}_{\gamma(t)} \left[ \left( \ln \left( \frac{\mathrm{d}\nu}{\mathrm{d}\mu} \right) - \mathbb{E}_{\gamma(t)} \left[ \ln \left( \frac{\mathrm{d}\nu}{\mathrm{d}\mu} \right) \right] \right)^2 \right] \mathrm{d}t.$$

The AM-GM inequality together with (30) for $k = 2$ justify differentiating under the integral sign, yielding the standard identity

$$\frac{\mathrm{d}}{\mathrm{d}t} \left( \mathbb{E}_{\gamma(t)} \left[ \ln \left( \frac{\mathrm{d}\nu}{\mathrm{d}\mu} \right) \right] \right) = \mathbb{E}_{\gamma(t)} \left[ \left( \ln \left( \frac{\mathrm{d}\nu}{\mathrm{d}\mu} \right) - \mathbb{E}_{\gamma(t)} \left[ \ln \left( \frac{\mathrm{d}\nu}{\mathrm{d}\mu} \right) \right] \right)^2 \right].$$

This, together with the application of integration by parts gives us

$$\begin{aligned}
D^{\mathrm{FR}}(\mu \mid \nu) &= \int_0^1 t \cdot \frac{\mathrm{d}}{\mathrm{d}t} \left( \mathbb{E}_{\gamma(t)} \left[ \ln \frac{\mathrm{d}\nu}{\mathrm{d}\mu} \right] \right) \mathrm{d}t \\
&= \mathbb{E}_{\gamma(1)} \left[ \ln \frac{\mathrm{d}\nu}{\mathrm{d}\mu} \right] - \int_0^1 \left( \mathbb{E}_{\gamma(t)} \left[ \ln \frac{\mathrm{d}\nu}{\mathrm{d}\mu} \right] \right) \mathrm{d}t.
\end{aligned} \tag{31}$$

Finally, using the fact that $\gamma(0) = \mu$ and $\gamma(1) = \nu$ and $\frac{\dot{Z}(t)}{Z(t)} = \int_{\mathbb{R}^n} \ln \left( \frac{\mathrm{d}\nu}{\mathrm{d}\mu} \right) \mathrm{d}\gamma(t)$, we get that

$$\int_0^1 \left( \mathbb{E}_{\gamma(t)} \left[ \ln \frac{\mathrm{d}\nu}{\mathrm{d}\mu} \right] \right) \mathrm{d}t = \int_0^1 \frac{\dot{Z}(t)}{Z(t)} \, \mathrm{d}t = \int_0^1 \frac{\mathrm{d}}{\mathrm{d}t} \ln(Z(t)) \, \mathrm{d}t = \ln(Z(1)) - \ln(Z(0)) = 0,$$

which finally gives us

$$D^{\mathrm{FR}}(\mu \mid \nu) = \mathbb{E}_\nu \left[ \ln \frac{\mathrm{d}\nu}{\mathrm{d}\mu} \right] = \int_{\mathbb{R}^n} \ln \left( \frac{\mathrm{d}\nu}{\mathrm{d}\mu} \right) \mathrm{d}\nu = \mathrm{KL}(\nu \mid \mu). \qquad \square$$

### B.3. The Stein metric

In this subsection, we introduce the Stein metric from SVGD and give a closed-form expression for the resulting divergence $D^{\mathrm{Stein}}$ between two centered elliptic distributions.

*Example* B.4 (Stein metric). Consider a symmetric positive definite kernel $K \colon \mathbb{R}^d \times \mathbb{R}^d \to \mathbb{R}$, for example, a heat kernel. The *Stein isomorphism* (Duncan et al., 2023; Liu & Wang, 2016; Liu, 2017; Nüsken & Renger, 2023) at a density $\rho \in L^1(\mathbb{R})$ is given by

$$\varphi_\rho^{\mathrm{Stein}}([f]) := -\operatorname{div} \left( \rho \int_{\mathbb{R}^d} K(\cdot, y) \operatorname{grad}(f)(y) \rho(y) \, \mathrm{d}y \right).$$

The associated inner product on the cotangent space is

$$\langle\!\langle [f], [g] \rangle\!\rangle_\rho^{\mathrm{Stein}} := \int_{\mathbb{R}^d} \int_{\mathbb{R}^d} K(x, y) \nabla f(x)^\mathsf{T} \nabla g(y) \rho(x) \rho(y) \, \mathrm{d}x \, \mathrm{d}y.$$

By considering density-dependent kernels, one can also construct a "normalized Stein metric" (Xu et al., 2022). ●

From now on, we focus on the kernel $K(x, y) := x^\mathsf{T} A y + a$ for the Stein metric, where $A \in \mathrm{Sym}_+(\mathbb{R}; d)$ and $a \geq 0$. We choose this kernel because its simple structure facilitates closure in the family of elliptical distributions for other (accelerated) gradient flows (Liu et al., 2024; Stein & Li, 2026; Stein et al., 2026a).

**Lemma B.5** (Stein vector field with generalized bilinear kernel and elliptic distribution). *Let $\rho_{m,\Sigma,g} \sim E(m, \Sigma, g)$ and $\nabla f(x) = Bx + b$. Then, for $x \in \mathbb{R}^d$ we have*

$$v^{\mathrm{Stein}}(\rho, \nabla f)(x) = \int_{\mathbb{R}^d} (x^\mathsf{T} A y + a)(By + b) \rho_{m,\Sigma,g}(y) \, \mathrm{d}y = \left( B(\kappa_g \Sigma + m m^\mathsf{T}) + b m^\mathsf{T} \right) A x + a(Bm + b). \tag{32}$$

*Proof.* For an $E(m, \Sigma, g)$-distributed random variable $X$, the integral can be expressed as

$$\mathbb{E}[(x^T A X)(BX + b)] = \left( B \mathbb{E}[XX^\mathsf{T}] + b \mathbb{E}[X]^\mathsf{T} \right) Ax$$

By Lemma B.1, $\mathbb{E}[X] = m$ and

$$\mathbb{E}[XX^\mathsf{T}] = \mathbb{V}[X] + mm^\mathsf{T} = \kappa_g \Sigma + mm^\mathsf{T},$$

so that

$$\int_{\mathbb{R}^d} x^\mathsf{T} Ay(By + b) p_{m,\Sigma,g}(y)\, \mathrm{d}y = \left( B(\kappa_g \Sigma + mm^\mathsf{T}) + bm^\mathsf{T} \right) Ax.$$

The second summand is immediate. $\qquad\square$

*Remark* B.6. Recall that the KL between two Gaussians is

$$\mathrm{KL}(\mathcal{N}(m_0, \Sigma_0) \mid \mathcal{N}(m_1, \Sigma_1)) = \frac{1}{2} \left[ \Delta m^\mathsf{T} \Sigma_1^{-1} \Delta m - \mathrm{tr}\left( \Sigma_1^{-1} \Delta \Sigma \right) + \ln\left( \frac{\det(\Sigma_1)}{\det(\Sigma_0)} \right) \right], \tag{33}$$

where $\Delta m := m_1 - m_0$ and $\Delta \Sigma := \Sigma_1 - \Sigma_0$.

The Stein divergence with bilinear kernel between Gaussian distributions corresponds to a "pre-conditioned" (reverse) KL divergence (compare with (33)), so for this simple kernel, we do not get "state-space-awareness".

**Theorem B.7** (Stein divergence with bilinear kernel)**.** *Suppose $\mu \sim \mathcal{N}(0, \Sigma_0)$ and $\nu \sim \mathcal{N}(0, \Sigma_1)$ are centered Gaussians, $K(x,y) = x^\mathsf{T} Ay + a$, and $\Sigma_0, \Sigma_1$ and $A$ all commute. Then with $R = \Sigma_1 \Sigma_0^{-1}$ we have*

$$D^{\mathrm{Stein}}(\mu \mid \nu) = \frac{1}{4} \mathrm{tr}\left( A^{-1}(R - I_d - \log(R)) \right).$$

*In particular, if $A = I_d$, then $D^{\mathrm{Stein}}(\mu \mid \nu) = \frac{1}{2} \mathrm{KL}(\nu \mid \mu)$.*

*Proof.* Setting $b = 0$ in (10) and plugging in the Stein vector field (32) yields

$$\begin{cases} \dot{m}_t = B(\kappa_g \Sigma_t + m_t m_t^\mathsf{T}) A m_t, \\ \dot{\Sigma}_t = 2\,\mathrm{Sym}\left( \Sigma_t B(\kappa_g \Sigma_t + m_t m_t^\mathsf{T}) A \right). \end{cases}$$

First, $m_0 = m_1 = 0$ implies that $\dot{m}_t = m_t = 0$, so we only have to focus on the covariance equation.

Since $A, \Sigma_0, \Sigma_1$ are simultaneously diagonalizable, we can work in this basis to reduce this ODE system to

$$\dot{\sigma}_i(t) = 2\beta_i \kappa_g \sigma_i(t)^2 a_i.$$

where for $i \in \{1, \dots, d\}$ and $t \geq 0$, $a_i > 0$, $\sigma_i(t) > 0$, and $\beta_i$ are the eigenvalues of $A$, $\Sigma_t$ and $B$, respectively. The solution is given by

$$\sigma_i(t) = \frac{1}{\sigma_i(0)^{-1} - 2\kappa_g a_i \beta_i t}, \qquad i \in \{1, \dots, d\}.$$

Solving for $\beta_i$ in terms of the boundary conditions yields

$$\beta_i = \frac{1}{2\kappa_g a_i} \left( \sigma_i(0)^{-1} - \sigma_i(1)^{-1} \right), \qquad i \in \{1, \dots, d\},$$

which can be written in matrix form as

$$B = \frac{1}{2\kappa_g} A^{-1} \left( \Sigma_0^{-1} - \Sigma_1^{-1} \right)$$

This also shows a posteriori that the solution for $\sigma_i$ exists on $[0, 1]$, so everything is well-defined. By (11),

$$D^{\mathrm{Stein}}(\mu \mid \nu) = \frac{\kappa_g}{2} \left( \mathrm{tr}(B\Sigma_1) - \int_0^1 \mathrm{tr}(B\Sigma_t)\, \mathrm{d}t \right) = \frac{\kappa_g}{2} \sum_{i=1}^d \beta_i \left( \sigma_i(1) - \int_0^1 \sigma_i(t)\, \mathrm{d}t \right)$$

$$= \sum_{i=1}^d \frac{1}{4a_i} \left( \frac{\sigma_i(1)}{\sigma_i(0)} - 1 - \ln\left( \frac{\sigma_i(1)}{\sigma_i(0)} \right) \right). \qquad\square$$

We leave the exploration of the Stein divergence for different kernels for future work.

## C. Linear Quadratic Optimal Control with Discounted Cost

This appendix provides a concise, self-contained statement and proof of the Bellman optimality equation along with its solution for the discounted infinite-horizon linear–quadratic regulator problem.

Let the (measurable) state space be $\mathcal{X} \subseteq \mathbb{R}^n$ and the (measurable) action space be $\mathcal{U} \subseteq \mathbb{R}^m$. We consider controlled Markov dynamics

$$x_{k+1} = f(x_k, u_k, w_k), \tag{34}$$

where $w_k$ are i.i.d. random disturbances (on a probability space $(\Omega, \mathcal{F}, \mathbb{P})$), and the initial state $x_0$ satisfies $\mathbb{E}[x_0] = \mu_0$ and $\mathbb{E}[(x_0 - \mu_0)(x_0 - \mu_0)^\top] = \Sigma_0$. A memory-less (static) policy $\pi = (\pi_0, \pi_1, \dots)$ is a sequence of measurable maps $\pi_k : \mathcal{X} \to \mathcal{U}$ and the discounted cost under policy $\pi$ is

$$J(\pi) := \mathbb{E}\left[\sum_{k=0}^{\infty} \gamma^k \ell(x_k, u_k)\right], \tag{35}$$

where $\ell : \mathcal{X} \times \mathcal{U} \to [0, \infty)$ is the stage cost, $u_k = \pi_k(x_k)$ and $\gamma \in (0, 1)$ is the discount factor. For a given policy $\pi$, define the value function $V^\pi$ as

$$V^\pi(x) := \mathbb{E}\left[\sum_{k=0}^{\infty} \gamma^k \ell(x_k, u_k) \,\Big|\, x_0 = x\right] \tag{36}$$

and the optimal value function

$$V^*(x) := \inf_\pi V^\pi(x), \qquad x \in \mathcal{X}. \tag{37}$$

Note that

$$J(\pi) = \mathbb{E}\left[\sum_{k=0}^{\infty} \gamma^k \ell(x_k, u_k)\right] = \mathbb{E}\left[\mathbb{E}\left[\sum_{k=0}^{\infty} \gamma^k \ell(x_k, u_k)\Big| x_0\right]\right] = \mathbb{E}\left[V^\pi(x_0)\right].$$

We assume the standard discounted LQR conditions: $(A, B)$ is stabilizable and $(A, Q^{1/2})$ is detectable, ensuring existence and uniqueness of optimal linear feedback policies for the cost defined below (Anderson & Moore, 2007). The following result is classical in dynamic programming and optimal control; see, for example, (Bertsekas, 2011).[7] We include a self-contained proof here, both to fix notation and because the result in the precise form and setting used in this paper is not readily available in the literature.

**Theorem C.1.** *Let $\mathcal{X} = \mathbb{R}^n$, $\mathcal{U} = \mathbb{R}^m$ and consider the problem of minimizing $J(\pi)$ given in* (35) *with quadratic stage cost*

$$\ell(x, u) = \frac{1}{2}\left(x^\top Q x + u^\top R u\right), \qquad Q \succeq 0, \; R \succ 0,$$

*discount factor $\gamma \in (0, 1)$ and linear Markovian dynamics*

$$x_{k+1} = A x_k + B u_k + w_k$$

*where $A \in \mathbb{R}^{n \times n}$ and $B \in \mathbb{R}^{n \times m}$. Assume that the initial state $x_0$ and the exogenous disturbances $w_k$ satisfy the following statistical properties:*

1. *$\mathbb{E}[x_0] = \mu_0$ and $\mathbb{E}[(x_0 - \mu_0)(x_0 - \mu_0)^\top] = \Sigma_0$ for $\Sigma_0 \succeq 0$ and*

2. *$w_k$ are i.i.d. with $\mathbb{E}[w_k] = 0$ and $\mathbb{E}[w_k w_k^\top] = \Sigma_w$ for $\Sigma_w \succeq 0$.*

*Consider the value function $V^\pi$ under a given policy $\pi$ as defined in* (36) *and the optimal value function $V^*$ as defined in* (37). *Assume that the pair $(A, B)$ is stabilizable and $(A, Q^{1/2})$ is detectable. Then the optimal value function $V^*$ satisfies the Bellman equation*

$$V^*(x) = \inf_{u \in \mathcal{U}} \left\{\ell(x, u) + \gamma \, \mathbb{E}\left[V^*(x_1) \mid x_0 = x, \; u_0 = u\right]\right\}, \qquad x \in \mathcal{X}. \tag{38}$$

---

[7]See also the expository blog post by Stephen Tu for an intuitive derivation: https://stephentu.github.io/blog/control-theory/optimal-control/2017/12/09/discounted-infinite-horizon-lqr.html.

*Furthermore, the quadratic optimal value function $V^*(x) = \frac{1}{2}\left(x^\top P x + \frac{\gamma}{1-\gamma}\operatorname{tr}(P\Sigma_w)\right)$ satisfies the Bellman equation* (38) *where $P$ is the unique positive semi-definite solution to the algebraic Riccati equation*

$$A_\gamma^\top P A_\gamma - P + Q - A_\gamma^\top P B (B^\top P B + R_\gamma)^{-1} B^\top P A_\gamma = 0, \tag{39}$$

*where $A_\gamma = \sqrt{\gamma}A$ and $R_\gamma = \frac{1}{\gamma}R$. The optimal cost is given by*

$$J^* = J(\pi^*) = \inf_\pi J(\pi) = E\left[V^*(x_0)\right],$$

*and the the optimal policy $\pi^*$ (time-invariant) is given by $\pi^* = (\pi_F, \pi_F, \cdots)$ with $\pi_F : \mathbb{R}^n \ni x \mapsto u = Fx \in \mathbb{R}^m$ where $F = -\left(R + \gamma B^\top P B\right)^{-1}\gamma B^\top P A.$*

*Proof.* Let us first note that the stabilizability of $(A, B)$ implies that there exists a matrix $F \in \mathbb{R}^{m \times n}$ such that the matrix $(A + BF)$ is Schur. This means that there exists a linear time-invariant policy $\pi = (\pi_F, \pi_F, \cdots)$ with $\pi_F : \mathbb{R}^n \ni x_k \mapsto u_k = Fx_k \in \mathbb{R}^m$ for all $k$, such that $V^\pi(x) < \infty$ for any $x \in \mathbb{R}^n$. Thus, the optimization problem is well-defined.

By using the tower property of conditional expectation when conditioning on $x_1$, we get that for any policy $\pi = (\pi_0, \pi_1, \pi_2, \dots)$,

$$\begin{aligned}
V^\pi(x) = \mathbb{E}\left[\sum_{k=0}^\infty \gamma^k \ell(x_k, u_k) \,\Big|\, x_0 = x\right] &= \ell(x, \pi_0(x)) + \gamma\mathbb{E}\left[\sum_{k=0}^\infty \gamma^k \ell(x_{k+1}, u_{k+1}) \,\Big|\, x_0 = x\right] \\
&= \ell(x, \pi_0(x)) + \gamma\mathbb{E}\left[\mathbb{E}\left[\sum_{k=0}^\infty \gamma^k \ell(x_{k+1}, u_{k+1})\Big|x_1\right] \,\Big|\, x_0 = x\right] \\
&= \ell(x, \pi_0(x)) + \gamma\mathbb{E}\left[V^{\pi^+}(x_1) \mid x_0 = x\right], \tag{40}
\end{aligned}$$

where $\pi^+ = (\pi_1, \pi_2, \pi_3, \cdots)$. We now prove the Bellman equation (38) by showing

$$\inf_{u \in \mathcal{U}}\{\ell(x, u) + \gamma\,\mathbb{E}[V^*(x_1) \mid x_0 = x, u_0 = u]\} \geq V^*(x) \geq \inf_{u \in \mathcal{U}}\{\ell(x, u) + \gamma\,\mathbb{E}[V^*(x_1) \mid x_0 = x, u_0 = u]\}. \tag{41}$$

Using (40) and the fact that $V^* \leq V^\pi$ pointwise for any $\pi$, we get that

$$V^\pi(x) = \ell(x, \pi_0(x)) + \gamma\,\mathbb{E}\left[V^{\pi^+}(x_1) \mid x_0 = x\right] \geq \ell(x, \pi_0(x)) + \gamma\,\mathbb{E}\left[V^*(x_1) \mid x_0 = x\right].$$

Taking the infimum over all policies (equivalently over the first-stage action $u = \pi_0(x)$ on the RHS) yields the lower bound

$$V^*(x) \geq \inf_{u \in \mathcal{U}}\{\ell(x, u) + \gamma\,\mathbb{E}[V^*(x_1) \mid x_0 = x, u_0 = u]\}. \tag{42}$$

We now prove the upper bound of $V^*$ in (41). By the definition of $V^*(x)$, we get that for any $x \in \mathcal{X}$ and any $\varepsilon > 0$, there exists a policy $\pi^{\varepsilon,x}$ such that

$$V^{\pi^{\varepsilon,x}}(x) \leq V^*(x) + \varepsilon.$$

We can now construct a new causal policy $\pi^\varepsilon = (\pi_0^\varepsilon, \pi_1^\varepsilon, \pi_2^\varepsilon, \cdots)$, where $\pi_0^\varepsilon : \mathcal{X} \to \mathcal{U}$ is an arbitrary map and $(\pi_1^\varepsilon, \pi_2^\varepsilon, \cdots) = \pi^{\varepsilon,x_1}$ which depends on the realization $x_1$. Using (40), we get that

$$\begin{aligned}
V^*(x) \leq V^{\pi^\varepsilon}(x) &= \ell(x, \pi_0^\varepsilon(x)) + \gamma\,\mathbb{E}\left[V^{(\pi^\varepsilon)^+}(x_1) \mid x_0 = x\right] \\
&= \ell(x, \pi_0^\varepsilon(x)) + \gamma\,\mathbb{E}\left[V^{\pi^{\varepsilon,x_1}}(x_1) \mid x_0 = x\right] \\
&\leq \ell(x, \pi_0^\varepsilon(x)) + \gamma\,\mathbb{E}\left[V^*(x_1) \mid x_0 = x\right] + \gamma\varepsilon.
\end{aligned}$$

Since the above inequality holds for any map $\pi_0^\varepsilon$ and arbitrary $\varepsilon > 0$, we can optimize over maps $\pi_0^\varepsilon$ (effectively over $u \in \mathcal{U}$) and take the limit as $\varepsilon$ goes to $0$ to obtain

$$V^*(x) \leq \inf_{u \in \mathcal{U}}\left(\ell(x, u) + \gamma\,\mathbb{E}\left[V^*(x_1) \mid x_0 = x, u_0 = u\right]\right). \tag{43}$$

This completes the proof of (41), which proves (38).

We now plug in the quadratic form of $V^* = \frac{1}{2}\left(x^\top P x + c\right)$, along with the quadratic form of the stage cost $\ell$ as well as the expression for $x_1$ obtained from the system dynamics (34) into the Bellman equation (38) to get

$$x^\top P x + c = \inf_{u \in \mathcal{U}} \left( \begin{bmatrix} x^\top & u^\top \end{bmatrix} \begin{bmatrix} \gamma A^\top P A + Q & \gamma A^\top P B \\ \gamma B^\top P A & \gamma B^\top P B + R \end{bmatrix} \begin{bmatrix} x \\ u \end{bmatrix} \right) + \gamma \operatorname{tr}(P\Sigma_w) + \gamma c \tag{44}$$

The optimization problem over $\mathcal{U}$ can be solved to obtain the optimal $u^* = -\overbrace{\left(R + \gamma B^\top P B\right)^{-1} \gamma B^\top P A}^{F} x$ which gives us the optimal policy $\pi^*$ such that $V^{\pi^*} = V^*$. Plugging this back into equation (44) gives the Riccati equation (39) and $c = \frac{\gamma}{1-\gamma} \operatorname{tr}(P\Sigma_w)$. Under stabilizability and detectability assumptions, the Riccati equation has a unique positive semi-definite solution (Anderson & Moore, 2007). Finally, note that $J(\pi) = \mathbb{E}[V^\pi(x_0)]$. Since $V^\pi(x) \geq V^*(x)$ for any $x$ and for any $\pi$, we get that $\inf_\pi J(\pi) = \inf_\pi \mathbb{E}[V^\pi(x_0)] \geq \mathbb{E}[V^*(x_0)]$. On the other hand, since there exists a policy $\pi^*$ such that $V^{\pi^*} = V^*$, we get that $\mathbb{E}[V^*(x_0)] = \mathbb{E}[V^{\pi^*}(x_0)] \geq \inf_\pi \mathbb{E}[V^\pi(x_0)] = \inf_\pi J(\pi)$. This shows that $\inf_\pi J(\pi) = \mathbb{E}[V^*(x_0)]$.

$\square$

### C.1. Proof of Corollary 5.1

*Proof.* First note that since $B$ has full column rank, $R_\circ \succ 0$ for $\circ \in \{\mathrm{FR}, \mathrm{O}, \mathrm{KW}\}$. The first part of the statement is a straightforward application of Theorem C.1. Now assume $\Sigma_w = \rho I$ and $\lambda = 1$. With this choice, we get that,

$$R_{\mathrm{KW}} = \frac{1}{\rho+1} B^\top B \to B^\top B = R_{\mathrm{O}} \qquad\qquad \text{as } \rho \to 0 \quad \text{and}$$

$$\|R_{\mathrm{KW}} - R_{\mathrm{FR}}\| = \frac{1}{\rho(\rho+1)} \|B^\top B\| \to 0 \qquad\qquad \text{as } \rho \to \infty.$$

Using the perturbation analysis of DARE (see (Sun, 1998)), we get the limiting situations 1. and 2. Finally, if the spectral radius of $A$ is less than $\frac{1}{\sqrt{\gamma}}$, then the spectral radius of $A_\gamma = \sqrt{\gamma} A$ is less than 1. This implies that the Lyapunov equation

$$A_\gamma^\top P A_\gamma - P + Q = 0 \tag{45}$$

has a positive definite solution. Since $R_{\mathrm{FR}}$ grows unbounded as $\rho \to 0$, the term $(B^\top P_{\mathrm{FR}} B + R_\gamma)^{-1}$ approaches 0 and the solution of the Riccati equation (39) converges to the solution of the Lyapunov equation (45), and thus remains bounded. Therefore, the optimal gain $F_{KL}$ given by (16) converges to 0 as $\rho \to 0$. This completes the proof. $\square$

### C.2. Counter example showing the lack of decomposition into stage-wise costs

Consider the scalar controlled dynamical system

$$x_{t+1} = x_t + u_t + v_t, \qquad x_0 = 0, \tag{46}$$

where $v_1, v_2 \overset{\text{iid}}{\sim} \mathcal{N}(0,1)$. For a control sequence $u = (u_1, u_2)$, the resulting trajectory $(x_1, x_2)$ satisfies

$$x_1 = u_1 + v_1, \tag{47}$$
$$x_2 = x_1 + u_2 + v_2. \tag{48}$$

Hence the joint law of $(x_1, x_2)$ under control $u$ is Gaussian with mean and covariance

$$p_u(x_1, x_2) = \mathcal{N}\left( \begin{bmatrix} u_1 \\ u_1 + u_2 \end{bmatrix}, \begin{bmatrix} 1 & 1 \\ 1 & 2 \end{bmatrix} \right). \tag{49}$$

Equivalently, the density can be written as

$$p_u(x_1, x_2) = \frac{1}{2\pi} \exp\left( -\frac{1}{2}\left[ (x_1 - u_1)^2 + (x_2 - x_1 - u_2)^2 \right] \right). \tag{50}$$

For the zero-control reference trajectory $u = (0, 0)$, we obtain

$$q(x_1, x_2) = \mathcal{N}\left(\begin{bmatrix} 0 \\ 0 \end{bmatrix}, \begin{bmatrix} 1 & 1 \\ 1 & 2 \end{bmatrix}\right), \tag{51}$$

or equivalently

$$q(x_1, x_2) = \frac{1}{2\pi} \exp\left(-\frac{1}{2}\left[x_1^2 + (x_2 - x_1)^2\right]\right). \tag{52}$$

The corresponding marginal distributions are

$$p_u(x_1) = \mathcal{N}(u_1, 1), \qquad\qquad q(x_1) = \mathcal{N}(0, 1), \tag{53}$$

that is,

$$p_u(x_1) = \frac{1}{\sqrt{2\pi}} \exp\left(-\frac{(x_1 - u_1)^2}{2}\right), \qquad\qquad q(x_1) = \frac{1}{\sqrt{2\pi}} \exp\left(-\frac{x_1^2}{2}\right). \tag{54}$$

Moreover, the one-step conditional laws are

$$p_u(x_2 \mid x_1) = \mathcal{N}(x_1 + u_2, 1), \qquad\qquad q(x_2 \mid x_1) = \mathcal{N}(x_1, 1), \tag{55}$$

or explicitly

$$p_u(x_2 \mid x_1) = \frac{1}{\sqrt{2\pi}} \exp\left(-\frac{(x_2 - x_1 - u_2)^2}{2}\right), \tag{56}$$

$$q(x_2 \mid x_1) = \frac{1}{\sqrt{2\pi}} \exp\left(-\frac{(x_2 - x_1)^2}{2}\right). \tag{57}$$

Note that

$$D^{\mathrm{WKL}}\left(p_u(x_1, x_2) \,\|\, q(x_1, x_2)\right) = \frac{1}{2}\left(2u_1^2 + u_2^2 + 2u_1 u_2\right)$$

$$\neq \frac{1}{2}\left(u_1^2 + u_2^2\right) = D^{\mathrm{WKL}}\left((p_u(x_1) \,\|\, q(x_1)\right) + D^{\mathrm{WKL}}\left((p_u(x_2 \mid x_1) \,\|\, q(x_2 \mid x_1)\right).$$

One can show this similarly for $D^{\mathrm{KW}}$. In contrast we have that

$$D^{\mathrm{KL}}\left(p_u(x_1, x_2) \,\|\, q(x_1, x_2)\right) = D^{\mathrm{KL}}\left((p_u(x_1) \,\|\, q(x_1)\right) + D^{\mathrm{KL}}\left((p_u(x_2 \mid x_1) \,\|\, q(x_2 \mid x_1)\right).$$

### C.3. Matrices introduced in Equation (61)

$$\mathcal{A} = \begin{bmatrix} A \\ A^2 \\ \vdots \\ A^{T-1} \end{bmatrix}, \quad \mathcal{B}_u = \begin{bmatrix} B & 0 & \cdots & 0 \\ AB & B & \ddots & \vdots \\ \vdots & \ddots & \ddots & 0 \\ A^{T-2}B & \cdots & AB & B \end{bmatrix}, \quad \mathcal{B}_w = \begin{bmatrix} I & 0 & \cdots & 0 \\ A & I & \ddots & \vdots \\ \vdots & \ddots & \ddots & 0 \\ A^{T-2} & \cdots & A & I \end{bmatrix}. \tag{58}$$

### C.4. Control penalty matrices under path-integral regularization

$$\mathcal{R}_{\mathrm{FR}} = \mathcal{B}_u^{\mathsf{T}}\left(\mathcal{B}_w(I \otimes \Sigma_w)\mathcal{B}_w^{\mathsf{T}}\right)^{-1}\mathcal{B}_u, \quad \mathcal{R}_{\mathrm{O}} = \mathcal{B}_u^{\mathsf{T}}\mathcal{B}_u, \quad \mathcal{R}_{\mathrm{KW}} = \mathcal{B}_u^{\mathsf{T}}\left(\mathcal{B}_w(I \otimes \Sigma_w)\mathcal{B}_w^{\mathsf{T}} + \lambda I\right)^{-1}\mathcal{B}_u. \tag{59}$$

## C.5. Optimal control with path-integral formulation

**Theorem C.2** (Optimal control under path-integral regularization)**.** *Consider the LTI system* (14) *with fixed initial state* $x_0 \in \mathbb{R}^n$, *and i.i.d. process noise* $w_t \sim \mathcal{N}(0, \Sigma_w)$. *Let* $\lambda > 0$ *and* $\mathcal{R}_\circ$ *be as defined in* (59) *for* $\circ \in \{\mathrm{FR}, \mathrm{O}, \mathrm{KW}\}$. *The optimal control sequence* $u_\circ$ *minimizing* $V^\circ$ *is*

$$u_\circ = -(\mathcal{R}_\circ + \mathcal{B}_u^\top \mathcal{Q} \mathcal{B}_u)^{-1} \mathcal{B}_u^\top \mathcal{Q} \mathcal{A} x_0. \tag{60}$$

*In particular,* $u_\mathrm{O}$ *does not depend on* $\Sigma_w$. *If* $\Sigma_w = \rho I$ *for some* $\rho > 0$ *and* $\lambda = 1$, *then*

1. $\lim_{\rho \searrow 0} u_\mathrm{KW} = u_\mathrm{O}$.

2. $\lim_{\rho \to \infty} \|u_\mathrm{KW} - u_\mathrm{FR}\| = 0$.

3. $\lim_{\rho \searrow 0} u_\mathrm{FR} = 0$ *while* $\lim_{\rho \searrow 0} u_\mathrm{KW} = u_\mathrm{O}$ *is not zero in general.*

*Proof.* Substituting the stacked dynamics

$$x = \mathcal{A} x_0 + \mathcal{B}_u u + \mathcal{B}_w w, \tag{61}$$

into the objective yields

$$\begin{aligned}
V^\circ(u) &= \mathbb{E}_{w_t \sim \mathcal{N}(0, \Sigma_w)} \left[ \frac{1}{2} (\mathcal{A} x_0 + \mathcal{B}_u u + \mathcal{B}_w w)^\top \mathcal{Q} (\mathcal{A} x_0 + \mathcal{B}_u u + \mathcal{B}_w w) \right] + \frac{1}{2} u^\top \mathcal{R}_\circ u \\
&= \frac{1}{2} (\mathcal{A} x_0 + \mathcal{B}_u u)^\top \mathcal{Q} (\mathcal{A} x_0 + \mathcal{B}_u u) + \frac{1}{2} \operatorname{tr}(\mathcal{Q} \mathcal{B}_w (I \otimes \Sigma_w) \mathcal{B}_w^\top) + \frac{1}{2} u^\top \mathcal{R}_\circ u \\
&= \frac{1}{2} u^\top (\mathcal{R}_\circ + \mathcal{B}_u^\top \mathcal{Q} \mathcal{B}_u) u + u^\top \mathcal{B}_u^\top \mathcal{Q} \mathcal{A} x_0 + \frac{1}{2} \operatorname{tr}(\mathcal{Q} \mathcal{B}_w (I \otimes \Sigma_w) \mathcal{B}_w^\top).
\end{aligned}$$

Since $\mathcal{R}_\circ + \mathcal{B}_u^\top \mathcal{Q} \mathcal{B}_u$ is positive definite, $V^\circ$ is strictly convex and possesses the unique solution $u_\circ$ obtained by solving the first-order necessary conditions of optimality

$$(\mathcal{R}_\circ + \mathcal{B}_u^\top \mathcal{Q} \mathcal{B}_u) u_\circ + \mathcal{B}_u^\top \mathcal{Q} \mathcal{A} x_0 = 0$$

which gives the optimal solution (60). Now assume $\Sigma_w = \rho I$ and $\lambda = 1$. With this choice, we get that,

$$\mathcal{R}_\mathrm{KW} = \mathcal{B}_u^\top (\rho \mathcal{B}_w \mathcal{B}_w^\top + I)^{-1} \mathcal{B}_u \to \mathcal{B}_u^\top \mathcal{B}_u = \mathcal{R}_\mathrm{O} \qquad \text{as } \rho \to 0 \quad \text{and}$$

$$\|\mathcal{R}_\mathrm{KW} - \mathcal{R}_\mathrm{FR}\| = \|\mathcal{B}_u^\top \left( (\rho \mathcal{B}_w \mathcal{B}_w^\top + I)^{-1} - (\rho \mathcal{B}_w \mathcal{B}_w^\top)^{-1} \right) \mathcal{B}_u\| \to 0 \qquad \text{as } \rho \to \infty.$$

This proves the first two items. Finally, $\mathcal{R}_\mathrm{FR}$ grows unbounded as $\rho$ approaches $0$ which implies that $u_\mathrm{FR}$ converges to $0$. $\qquad\square$

## C.6. Results on the cart-pole system

### C.6.1. CART–POLE MODEL, LINEARIZATION AND DISCRETIZATION

The cart-pole system is depicted in Figure 5. The equations of motion are given by

$$(m_c + m_p) \ddot{x} + m_p l \ddot{\theta} \cos(\theta) - m_p l \dot{\theta}^2 \sin(\theta) = u,$$

$$m_p l \ddot{x} \cos(\theta) + m_p l^2 \ddot{\theta} + m_p g l \sin(\theta) = 0$$

where $x$ is the cart horizontal position, $\theta$ is the pendulum angle, $m_c$ is the cart mass, $m_p$ is the pendulum mass, $l$ is the pendulum length (distance from the pivot to the pendulum center of mass), $g$ is the coefficient of gravity, and $u$ is the horizontal force applied to the cart. Introducing the generalized coordinate $q = \begin{bmatrix} x & \dot{x} & \theta & \dot{\theta} \end{bmatrix}$, the dynamics can be written compactly as

$$\frac{dq}{dt} = f(q, u).$$

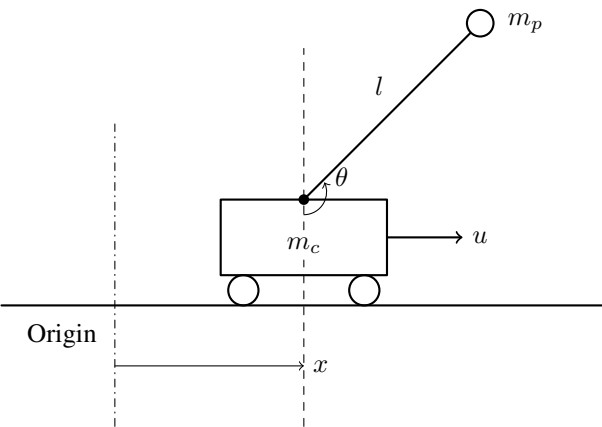

*Figure 5.* Cart-pole system

Consider a forced equilibrium $(q^*, u^*)$ such that $f(q^*, u^*) = 0$. Using the linear approximation of $f$ around this equilibrium, the non-linear system can be linearized to obtain an approximate linear time-invariant system in deviation variables $\tilde{q} = q - q^*$ as

$$\frac{d\tilde{q}}{dt} = \frac{dq}{dt} = f(q, u) \approx \underbrace{f(q^*, u^*)}_{0} + \underbrace{\frac{\partial f}{\partial q}(q^*, u^*)}_{A_c} \underbrace{(q - q^*)}_{\tilde{q}} + \underbrace{\frac{\partial f}{\partial u}(q^*, u^*)}_{B_c} \underbrace{(u - u^*)}_{\tilde{u}} = A_c \tilde{q} + B_c \tilde{u}.$$

For the upright equilibrium $q^* = \begin{bmatrix} 0 & 0 & \pi & 0 \end{bmatrix}$ and $u^* = 0$, we get

$$A_c = \begin{bmatrix} 0 & 1 & 0 & 0 \\ 0 & 0 & \dfrac{m_p g}{m_c} & 0 \\ 0 & 0 & 0 & 1 \\ 0 & 0 & \dfrac{(m_c + m_p)g}{l m_c} & 0 \end{bmatrix}, \qquad B_c = \begin{bmatrix} 0 \\ \dfrac{1}{m_c} \\ 0 \\ \dfrac{1}{l m_c} \end{bmatrix}.$$

Finally, we discretize the linearized continuous-time system with sampling time $T_s$ using a zero-order-hold scheme to obtain the discrete-time system

$$\tilde{q}_{k+1} = \underbrace{e^{A_c T_s}}_{A} \tilde{q}_k + \underbrace{\int_0^{T_s} e^{A_c \tau} B_c \, d\tau}_{B} \tilde{u}_k = A \tilde{q}_k + B \tilde{u}. \tag{62}$$

This discrete-time linear system $(A, B)$ is then used as input to the LQR design.

### C.6.2. SIMULATION RESULTS WITH THE CART-POLE SYSTEM

Figure 6 shows the cart position trajectories for the nonlinear cart-pole system. As with the double integrator, KL-regularization produces large oscillations, while WKL and KW yield increasingly stable trajectories as $\rho$ decreases, resulting in superior closed-loop performance.

### C.7. Double integrator with anisotropic noise

We now consider a planar double-integrator system obtained by stacking two independent copies of the one-dimensional model in Section 5.2. The state is ordered as $x_t = [q_{1,t}, \, p_{1,t}, \, q_{2,t}, \, p_{2,t}]^\top$ and the control as $u_t = [u_{1,t}, \, u_{2,t}]^\top$, so that the two spatial dimensions are decoupled. We set

$$A = I_2 \otimes \begin{bmatrix} 1 & 1 \\ 0 & 1 \end{bmatrix}, \qquad B = I_2 \otimes \begin{bmatrix} 0 \\ 1 \end{bmatrix}.$$

As in the one-dimensional case, we use $Q = I_4$ and $\gamma = 0.9$.

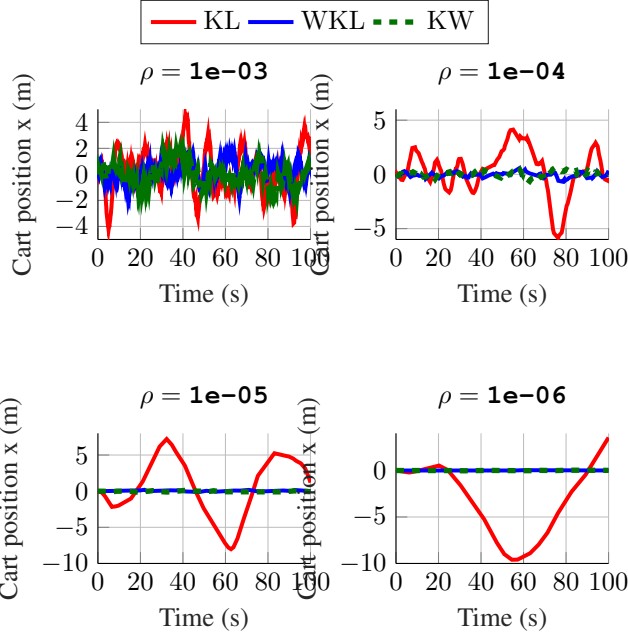

*Figure 6.* Closed-loop cart positions for the nonlinear cart–pole under KL-, WKL-, and KW-regularized controllers for $\rho \in \{10^{-3}, 10^{-4}, 10^{-5}, 10^{-6}\}$. At low noise, KL produces near-zero feedback and large oscillations, whereas WKL and KW reduce oscillations.

To isolate the effect of anisotropic process noise, we consider

$$w_t \sim \mathcal{N}(0, \Sigma_w^{(0)}), \qquad \Sigma_w^{(0)} = \mathrm{diag}(10^{-5}, 10^{-5}, 10^1, 10^1),$$

so that the first spatial dimension has very small noise, while the second has very large noise. We compute the optimal discounted LQR gains for the KL-, WKL-, and KWKL-regularized controllers, using $\lambda = 1$ in the KWKL case.

The resulting optimal feedback gains are

$$F_{\mathrm{FR}} = \begin{bmatrix} 0 & 0 & 0 & 0 \\ 0 & 0 & -0.5879 & -1.5875 \end{bmatrix},$$

$$F_{\mathrm{O}} = \begin{bmatrix} -0.3882 & -1.1817 & 0 & 0 \\ 0 & 0 & -0.3882 & -1.1817 \end{bmatrix},$$

$$F_{\mathrm{KW}} = \begin{bmatrix} -0.3882 & -1.1817 & 0 & 0 \\ 0 & 0 & -0.5879 & -1.5875 \end{bmatrix}.$$

This example highlights the directional behavior of the proposed regularizers. Since the two coordinates are decoupled, the optimal feedback gains remain block diagonal and the off-diagonal blocks are numerically zero. The three regularizers differ only in how they treat the two noise levels: KL suppresses control in the low-noise dimension while retaining nontrivial feedback in the high-noise dimension, WKL ignores the anisotropy and produces identical gains across both dimensions, and KWKL adapts direction-wise, matching WKL in the low-noise dimension and KL in the high-noise dimension. Thus, KWKL preserves the useful noise-dependent geometry where it is informative, while avoiding the low-noise degeneracy exhibited by KL.

### C.8. Optimal cost for the double integrator system from Section 5.2

Figure 7 shows the optimal costs with different regularizers, supporting the observations of Section 5.2

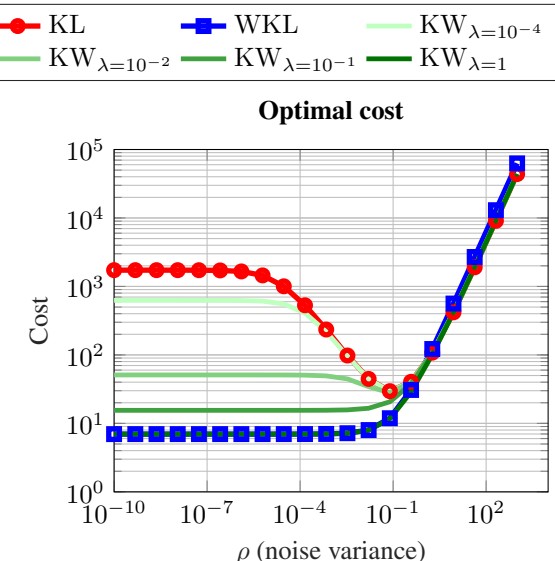

*Figure 7.* Optimal costs $J^\circ$ as a function of noise $\rho$ for $\lambda \in \{10^{-4}, 10^{-2}, 10^{-1}, 1\}$

## C.9. Effects of varying $\lambda$ for the double integrator system from Section 5.2

Figure 8 and Figure 9 supplement the Figure 3 and Figure 2, respectively, and plot additional trajectories with different values of $\lambda$.

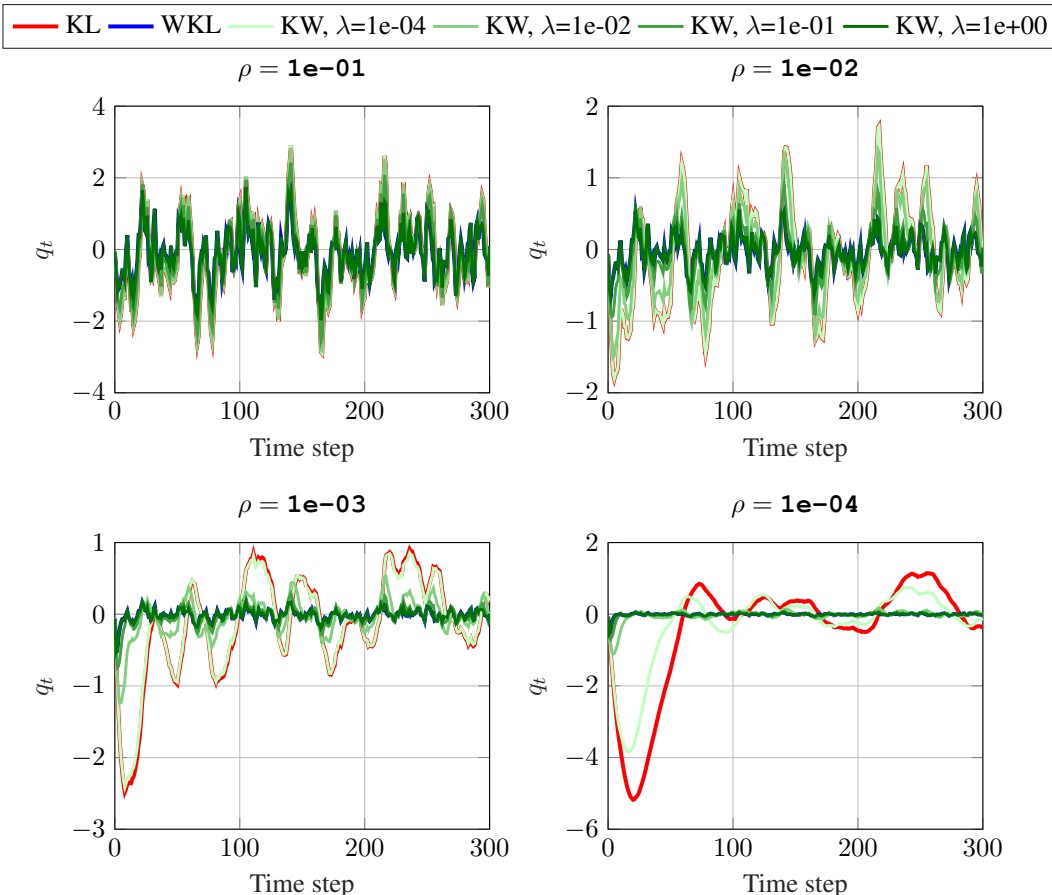

*Figure 8.* Closed-loop trajectories of the double integrator under KL-, WKL-, and KW-regularized controllers for noise levels $\rho \in \{10^{-1}, 10^{-2}, 10^{-3}, 10^{-4}\}$. At low noise, KL produces near-zero feedback and large oscillations, whereas WKL and KW yield more stable trajectories. For KW, varying the parameter $\lambda$ interpolates smoothly between KL-like and WKL-like behavior.

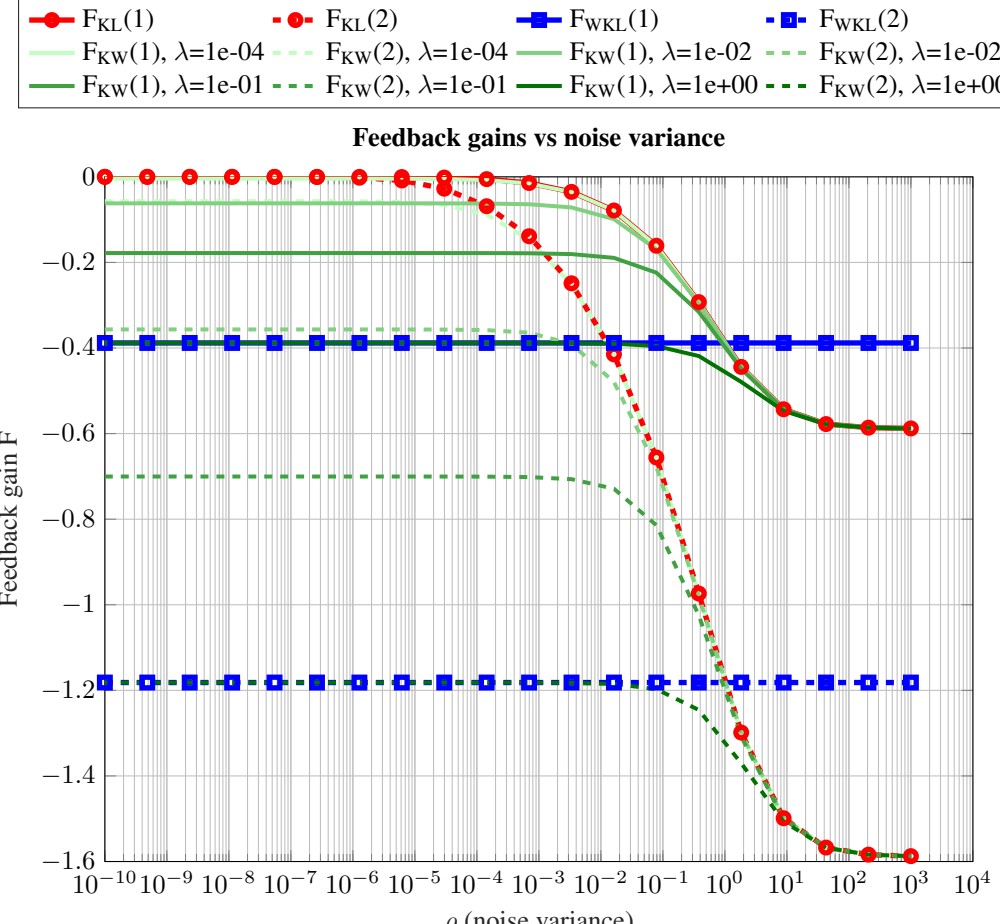

*Figure 9.* Comparison of optimal policies for $\{FR, O, KW\}$-regularized LQR. Each linear policy is represented by a $1 \times 2$ gain matrix $F$; both entries are shown. KL gains shrink to zero as noise disappears, WKL gains remain constant since they do not depend on $\rho$, and KW gains smoothly interpolate between the two.

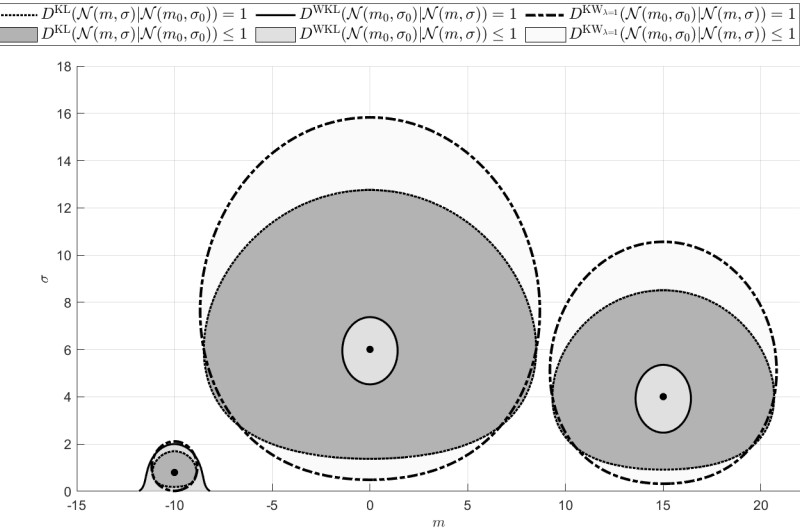

*Figure 10.* Comparison of sub-level sets for KL-divergence, WKL-divergence, and KWKL-divergence centered at various univariate Gaussian reference distributions $\mathcal{N}(m_0, \sigma_0)$, indicated by solid markers. Each shaded region represents the set of distributions $\mathcal{N}(m, \sigma)$ whose divergence from the reference is less than or equal to 1.

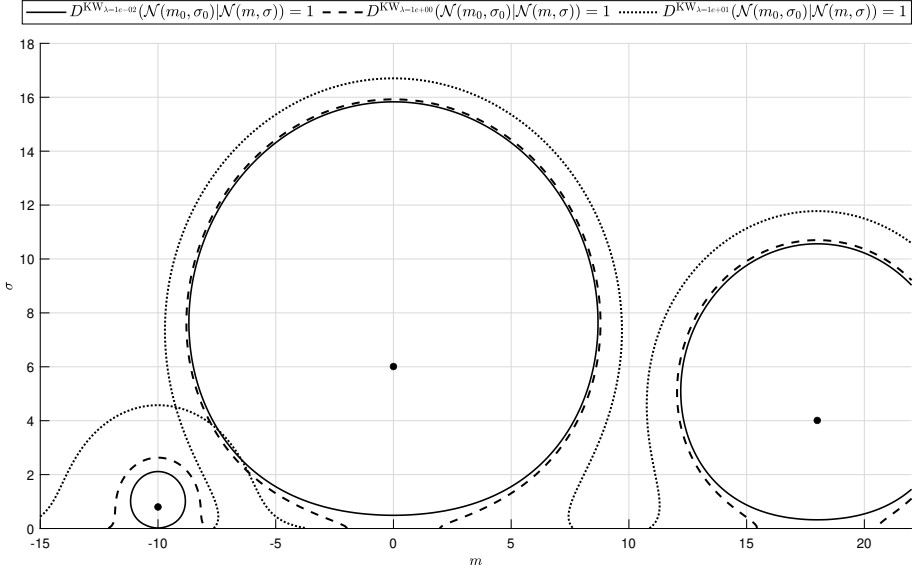

*Figure 11.* Level sets for KWKL-divergence centered at various univariate Gaussian reference distributions $\mathcal{N}(m_0, \sigma_0)$, indicated by solid markers for different values of $\lambda$.

### C.10. Comparison of divergence balls

Figure 10 compares sublevel sets of KL, WKL, and KWKL around univariate Gaussian reference distributions. Figure 11 shows how the KWKL level sets vary with the regularization parameter $\lambda$. Note that while the KL divergence balls are always convex, the WKL divergence balls and the KWKL divergence balls may become non-convex.

## D. Further Directions and Limitations

One shortcoming of the theory is that we can not offer any criteria ensuring the existence or uniqueness of geodesics. We conjecture that there exist two smooth elliptic densities with different generators that can not be connected by an O-dual

geodesic, but that there are also pairs of elliptic distributions with different generators that can be connected by an O-dual geodesic. If true, this would mean that in this geometry, the connected components of the density manifold are difficult to describe.

On the computational side, we acknowledge that outside of structured families, the proposed $G$-divergences generally require approximation. Possible approaches include sampling-based estimation of the corresponding geometric action, local Gaussian or elliptic approximations, moment-matching schemes, and plug-in approximations based on empirical covariance structure. Developing scalable estimators and optimization algorithms for nonlinear Markov decision processes is an important direction for future work.

Lastly, we plan to investigate new versions of the mutual information based on the divergences $D^G$ and apply the new regularizers to different problems, for example, in imaging, and to VAEs.

