# OpenReview forum: "Well-Posed KL-Regularized Control via Wasserstein and Kalman–Wasserstein KL Divergences"
_ICML.cc/2026/Conference — ICML 2026 regular_

### Official Review · Reviewer_1KbW · 2026-02-20

**Soundness:** 2
**Presentation:** 4
**Significance:** 2
**Originality:** 3
**Overall Recommendation:** 5
**Confidence:** 2

**Summary:**

This paper introduces a novel information-geometric framework that replaces the Fisher-Rao geometry in KL divergence with a transport-based geometry to address the instability and support-mismatch issues of traditional KL regularization in control and reinforcement learning.

**Compliance With Llm Reviewing Policy:**

Affirmed.

**Final Justification:**

The authors provide a clear and technically sound framework with solid theoretical derivations and supporting experiments, and the rebuttal adequately addresses my main questions regarding applicability and extensions. Overall, I find the work well-motivated and practically relevant, and I maintain a positive recommendation.

**Key Questions For Authors:**

1. Could the authors provide additional results on more complex experimental setups (e.g., standard RL benchmarks or higher-dimensional control tasks) to demonstrate the stability of the method in practical scenarios?

2. The current framework focuses on discrete-time linear systems, can this methodology be naturally extended to continuous-time linear systems? The author only need to give some intuitions, do not need rigorious proof or details.

**Limitations:**

yes

**Strengths And Weaknesses:**

Strengths

1: This paper presents details mathematical derivations and comprehensive proofs.

2: The manuscript is well-structured and the transition from classical information geometry to the proposed framework is easy to follow.

3: The authors provide experimental results that validate the theoretical claims within the specified settings.

Weaknesses

1: The experimental settings are relatively simple(2 dimension system or Cart-pole system), leaving it unclear whether the proposed method maintains its advantages in high-dimensional or complex environments.

2: The conceptual transition from KL to WKL/KW feels like a natural extension of existing optimal transport literature, which may limit the perceived novelty of the contribution.

---

> ### Author Rebuttal · Authors · 2026-03-31
>
> # Regarding the reviewer's first question on more complex experiments
> We thank the reviewer for this suggestion and agree that evaluating on more complex benchmarks would be valuable. At the same time, we would like to emphasize that our cart-pole experiments are already not purely linear: while the controllers are derived from a linearized model, they are evaluated on the full non-linear cart-pole dynamics. This demonstrates that the proposed regularizer can be effective beyond a purely linear setting.
> More broadly, the primary goal of this paper is to establish the divergence itself and show that it yields a well-posed control objective. In this regard, the closed-form structure of WKL/KWKL is particularly important, as it makes the regularizer computationally tractable and, in principle, scalable to higher-dimensional settings.
> We agree, however, that extending the empirical evaluation to standard RL benchmarks and more complex control tasks is an important direction for future work. As discussed in our response to Reviewer hi3G, the present results provide a foundation, and we give a roadmap for such extensions, including integration into existing RL pipelines and implementations in more general settings.
>
> We now describe a new experiment featuring anisotropic noise that was already alluded to in the rebuttal to reviewer rZH8:
> In this setup, the dynamics are decoupled across the two spatial dimensions, and we consider **low noise in the first dimension** and **high noise in the second**. The state is ordered as $[x, \dot{x}, y, \dot{y}]$, and the control as $[u_x, u_y]$. The resulting optimal feedback gains for $\\lambda=1$ are:
> $$F_{KL}=\begin{bmatrix}
>     0  &0  &0  &0 \\\\
>     0   &0  &-0.5879  &-1.5875
> \end{bmatrix}$$
> $$F_{WKL}=\begin{bmatrix}
>     -0.3882  &-1.1817  &0  &0 \\\\
>     0   &0  &-0.3882  &-1.1817
> \end{bmatrix}$$
> $$F_{KWKL}=\begin{bmatrix}
>     -0.3882  &-1.1817  &0  &0 \\\\
>     0   &0  &-0.5879  &-1.5875
> \end{bmatrix}$$
> As expected from the decoupled dynamics, the **off-diagonal blocks are (numerically) zero in all cases**, indicating that none of the regularizers introduces spurious cross-axis coupling. The differences between the methods are therefore isolated to how each treats the individual dimensions.
>
> This experiment highlights the distinct roles of WKL and KWKL:
>
> - **KL** suppresses control in the low-noise dimension (near-zero gains), while behaving normally in the high-noise dimension.
> - **WKL** ignores the anisotropy of the noise and produces identical gains across both dimensions.
> - **KWKL adapts direction-wise**:
>   - in the low-noise dimension, it matches WKL (avoiding KL’s degeneracy),
>   - in the high-noise dimension, it matches KL (preserving sensitivity to the noise structure).
>
> Thus, KWKL can be interpreted as **retaining useful noise-dependent geometry where it is informative, while reverting to a well-posed alternative where KL becomes pathological**. This directional adaptivity is not visible in isotropic settings and directly addresses the reviewer’s question regarding heteroscedastic noise.
>
> We will include this experiment and discussion in the revised manuscript to more clearly distinguish the roles of WKL and KWKL.
>
> # Regarding the first question on continuous-time setting
> Yes, we believe the framework can be extended naturally to continuous-time linear stochastic systems. At an intuitive level, one would replace the discrete-time sum of stage costs by a time integral, and replace the one-step transition distributions by the corresponding infinitesimal conditional laws or velocity/drift distributions. In the linear-Gaussian setting, this would lead to a continuous-time analogue of our regularization, which would also be quadratic in the control input. Once the optimal control problem is formulated, the well-known correspondence between discrete-time and continuous-time LQR formulations would lead to the same low-noise singularity that motivates our paper. We therefore view the continuous-time case as a natural analogue of the present framework rather than a fundamentally different problem.
>
> Lastly, we are not entirely sure how to interpret the second weakness the reviewer mentions. The novel transport-based divergences arise as a fusion of the KL divergence from information geometry and transport-based metrics on the space of probability densities, and can thus be seen as a generalization of divergence-like quantities in optimal transport.

---

> > ### Author Rebuttal · Reviewer_1KbW · 2026-04-01
> >
> > My concerns have been adequately addressed. Thank you. I will raise my score, while maintaining a relatively low confidence.

---

### Official Review · Reviewer_rZH8 · 2026-03-11

**Soundness:** 4
**Presentation:** 3
**Significance:** 3
**Originality:** 4
**Overall Recommendation:** 5
**Confidence:** 4

**Summary:**

This paper introduces two Wasserstein-geometry-based KL-type regularizers, WKL and KWKL, and develops closed-form analyses for them in elliptic-distribution and linear-control settings, with the goal of alleviating the ill-posedness of standard KL regularization in the low-noise limit.

**Compliance With Llm Reviewing Policy:**

Affirmed.

**Final Justification:**

I believe the underlying ideas are likely sound.

**Key Questions For Authors:**

1. **Regarding the relationship between this paper and general RL scenarios**

The paper uses policy search and trust-region methods as important motivation in the introduction, but the current theory and experiments mainly focus on analytically tractable LQR/LTI settings with Gaussian or more general elliptical distributions. Could the authors further clarify what the expected usage of the proposed $D_{\mathrm{WKL}}$ and $D_{\mathrm{KW}}$ would be in general nonlinear MDP or deep RL scenarios?

In particular, when the state distribution no longer remains within the Gaussian or elliptical distribution family after multi-step rollouts, do these two divergences still have extensible computational or approximation schemes, such as sampling-based estimation, local approximation, or other numerical implementations? The authors may also consider more explicitly delineating in the main text whether the current closed-form results and affine vector field assumptions are primarily intended to provide geometric analysis intuition, or can already be viewed as practical algorithmic preliminary steps toward broader RL/ML settings.


2. **Regarding the difference between WKL and KWKL in noise structure**


According to the derivations in the paper, the effective control penalty corresponding to WKL no longer explicitly depends on the process noise covariance, while KWKL retains partial dependence on the noise structure. Could the authors further discuss what practical implications this difference would have in heteroscedastic or anisotropic noise environments?

The current experiments mainly consider settings with scalar noise scaling, so I would also like to ask the authors to clarify: do these results primarily demonstrate that the proposed methods can avoid singularity in the low-noise regime, but have not yet sufficiently distinguished the difference between WKL and KWKL in terms of whether they preserve noise geometric information? If possible, supplementing with discussion or experiments under non-isotropic noise settings might better help understand the unique role of KWKL relative to WKL.

3. **Regarding the theoretical contribution levels of WKL and KWKL**


The main text mentions that some closed-form results for WKL can be viewed as natural generalizations of existing Gaussian results to more general elliptical distributions. Could the authors more explicitly distinguish the roles of WKL and KWKL in this paper in the introduction or contribution summary?


For example, is the more central theoretical increment of this paper mainly reflected in the construction of KWKL, its closed-form analysis, and its interpolation property in the low-noise limit, while WKL serves more as geometric motivation and a reference baseline? I believe that a clearer articulation of the positioning of both in terms of contribution hierarchy would help readers more accurately understand the theoretical main line of this paper.

**Limitations:**

Yes

**Strengths And Weaknesses:**

### Strengths




1.
The paper addresses the ill-posedness of standard KL penalties in low-noise or degenerate noise limits. The proposed WKL and KWKL remain well-posed and correspond to interpretable quadratic control costs, forming a coherent connection among problem setting, method design, and control applications.


2.
The paper goes beyond definitions to derive closed-form expressions for WKL and KWKL, explains KWKL's connection to covariance inflation, and establishes its interpolation property between KL and WKL. These results clarify the theoretical structure and enhance interpretability.






### Weaknesses




1. **The reliability of the core divergence formula derivation still requires further clarification.**
Theorem 4.1 and Appendix A.1 provide a closed-form expression for the linearized Wasserstein KL divergence with a leading coefficient of $3/2$. However, examining the derivation in the appendix, the authors first write the geodesic density evolution as $\rho_t = \rho_0 - t\Delta f$, but then perform symbol substitutions during the integration step that appear inconsistent with this evolution equation, making the final constant factor appear unstable. Given the canonical divergence definition provided in the paper and the constant-speed path structure under this flat geometry, readers would naturally expect the coefficient to be $1/2$ rather than $3/2$. Therefore, the closed-form result in Theorem 4.1 and its appendix proof exhibit a clear self-consistency issue at least at the level of the constant factor, which requires the authors to re-examine and clarify.




2. **The KW closed-form derivation relies on rather strong commutativity and diagonalizability structures, but this point is not emphasized early or clearly enough in the theoretical development.**
The covariance dynamics result in Lemma 4.8 is established under the premise of a linear velocity field $v_t(x) = S_t x + s_t$ with $S_t \in \mathrm{Sym}$. For the KW case, after writing the velocity field as $v_{KW}(x) = (\kappa_g \Sigma_t + \lambda I)(Bx + b)$, the corresponding matrix $S_t = (\kappa_g \Sigma_t + \lambda I)B$ is not automatically symmetric in general, so the subsequent closed-form analysis does not hold unconditionally. Although the authors further introduce simultaneously diagonalizable and commutative structures in Theorem 4.11 and its proof to close the derivation, this indicates that the KW theoretical results in this paper actually depend on a rather strong set of algebraic assumptions, not merely on elliptical distributions plus KW regularization alone. If the current manuscript does not delineate these structural prerequisites earlier and more explicitly, readers may overestimate the generality of the results.




3. **The existence conditions in Theorem 4.11 appear overly strong and significantly limit the theoretical coverage of the result.**
Theorem 4.11 imposes a coordinate-wise inequality constraint on the parameter $\lambda$:
$$\lambda < \frac{\kappa_g (s_0)_i (s_1)_i}{(s_1)_i - (s_0)_i},$$
but when the target covariance contracts in certain directions, i.e., $(s_1)_i < (s_0)_i$, the right-hand side becomes negative, while the paper consistently assumes $\lambda > 0$. This renders the theorem's conditions impossible to satisfy in many natural and important variance-shrinking scenarios, thereby significantly weakening the applicability of this result. Furthermore, the explicit solutions in the appendix indicate that the relevant denominator terms remain positive at the endpoints, so at least from how the current derivation is presented, this condition may not be necessary, or is at least stated too strongly. In other words, there remains tension between the well-posedness existence conditions currently given in Theorem 4.11 and the closed-form expressions in the appendix. The authors need to clarify more clearly whether this condition is a technical sufficient condition, a conservative bound used in the proof, or a genuinely necessary condition that could be further relaxed.

---

> ### Author Rebuttal · Authors · 2026-03-31
>
> We thank the reviewer for their comments and detailed reading of our submission.
> Concerning the weakness, we provide the following comments:
>
> * Regarding Theorem 4.1, the reviewer is precisely correct: we fixed a typo in the proof, and obtained the factor 1/2, which makes much more sense.
>
> * In Theorem 4.11, the condition on $\lambda$ is not needed. We will update the proof accordingly.
>
> * In Lemma 4.8, we do not need the assumption that $S_t$ is symmetric, and we will update accordingly.
> We will also generalize the statement and proof of Theorem 4.11 to centered distributions, where the commutativity assumption is not needed. In the general case, we can still derive an expression whose only non-closed form part is the fundamental matrix of the vector ODE $\dot{r}_{t} = (\Sigma_t + \lambda) r_{t}$, since the commutativity assumption ensures that the mean equation decouples into scalar equations.
>
> # Question regarding the relationship between this paper and general RL scenarios
> We thank the reviewer for raising this important point. Our current paper is not intended to present a full deep-RL algorithm, but rather to establish a well-posed divergence that can serve as a principled regularizer in RL/control settings. The LTI/LQR setting is chosen deliberately as a canonical and analytically tractable testbed in which the low-noise singularity can be isolated and studied rigorously. This is closely related to recent observations in KL-regularized RL from expert demonstrations, where a small predictive variance in the reference policy can cause the KL to blow up and lead to unstable optimization [1]. Although our setting considers KL between controlled and passive transition kernels rather than between policies, the same low-noise pathology arises, and the LTI setting provides a clean framework for precisely analyzing it.
>
> Moreover, the closed-form assumptions are less restrictive in practice than they may initially appear. In standard continuous-control RL, the actor is typically parameterized as a Gaussian policy with a neural network that parametrizes the mean and log-standard deviation, often with diagonal covariance; TRPO is a canonical example of this setup [2]. Thus, the Gaussian/Elliptic distributions and commutativity assumptions used in our analysis are already aligned with widely used policy classes. In particular, in actor-critic or trust-region methods, the KL term between Gaussian policies can be replaced by WKL/KWKL using the same mean/covariance parameterization, yielding a well-defined objective even when variances become small. This suggests that the proposed divergences are not only theoretically well-motivated but also directly compatible with existing RL pipelines.
>
> Regarding the reviewer’s question about non-Gaussian or non-elliptical state distributions after multi-step rollouts, we agree that the current paper does not provide a complete general-purpose estimator for our divergences. Our closed-form results are intended to provide the analytic core and geometric intuition; extending them to more general settings is an important direction for future work. A natural route would be to combine these divergences with sampling-based estimation. Developing estimators with strong statistical properties in such settings is an important open problem, but it goes beyond the scope of the present paper.
>
> # Regarding the difference between WKL and KWKL in noise structure
>
> Thank you for this insightful comment. We agree that the scalar-noise experiments primarily demonstrate the removal of the low-noise singularity and, by themselves, do not fully isolate the role of noise geometry. To address this, we performed an additional experiment with **anisotropic process noise** on a 2D double integrator, whose results are discussed in the rebuttal to reviewer 1KbW, since they raised a very similar issue.

---

> > ### Author Rebuttal · Reviewer_rZH8 · 2026-04-02
> >
> > The rebuttal is responsive and improves my confidence in the main claims. I will keep my postive rating.

---

### Official Review · Reviewer_hi3G · 2026-03-13

**Soundness:** 3
**Presentation:** 3
**Significance:** 2
**Originality:** 3
**Overall Recommendation:** 4
**Confidence:** 3

**Summary:**

This paper is motivated by the central issue of the KL divergence when used as a regularization term, particularly in settings where the support of probability measures becomes highly concentrated. Motivated by this issue, the paper introduces a new class of divergences between probability distributions, referred to as G-divergences.  The construction is based on a geodesic interpretation of divergences on the space of probability distributions. In particular, the classical KL divergence can be viewed as arising from the Fisher–Rao metric. The authors generalize this construction by replacing the Fisher–Rao metric with a more general metric G, which leads to the definition of the corresponding G-divergence.

The paper derives explicit formulas for the divergence for specific classes of distributions, enabling tractable computations in special settings. Moreover, the paper numerically analyzes the effect of these divergences when used as regularizers in an LQR optimal control problem. Through this example, the authors demonstrate how the proposed divergences lead to better-behaved regularization compared to the classical KL divergence, particularly in regimes where the latter becomes ill-posed.

**Compliance With Llm Reviewing Policy:**

Affirmed.

**Final Justification:**

The rebuttal raised my final judgement as it establishes theoretical properties of the metric, and points me to the remark that partially addresses the unique property of the proposed metric. The comparison to non KL metrics is missing.

**Key Questions For Authors:**

None.

**Limitations:**

To some extent.

**Strengths And Weaknesses:**

__Soundness__: Overall, I found the paper to be technically sound. The theoretical development is mathematically rigorous and the results appear correct to the extent that I checked them.

__Originality__:
- Conceptually, the geometric viewpoint is interesting and leads to a family of divergences that generalize KL. However, the geometric viewpoint on itself is not entirely novel, or at least the novelty does not come across to the reader.
- It is not entirely clear what new qualitative properties this divergence provides compared to existing alternatives that have already been studied extensively in the literature. The limitations of KL divergence—particularly its sensitivity to support mismatch—are well known and have motivated a large body of work introducing alternative divergences and distances, such as Wasserstein distances, integral probability metrics, maximum mean discrepancy (MMD), and other transport-based divergences. The paper would benefit from a clearer discussion of how the proposed divergence differs from or improves upon these existing alternatives.

__Signifiacne__:
- In my opinion, the significance and impact of the paper are currently somewhat limited. While the construction of the divergence is mathematically interesting, the paper does not clearly articulate what practical or conceptual advantages the new metric provides relative to existing divergences.
- The analysis carried out in specific settings—such as the LQR example—is well done, but the implications of these results are not sufficiently discussed.
- To better demonstrate the significance of the work, I believe the paper should go beyond comparisons with the classical KL divergence. Since many alternatives to KL already exist, it would be more convincing to compare the proposed metric with other modern divergences. This could be done either theoretically (e.g., by characterizing properties that distinguish the metric) or empirically (e.g., through numerical experiments illustrating advantages in certain regimes). For a paper that introduces a new divergence at a fairly fundamental level, a deeper analysis of when and why practitioners should prefer this metric over existing ones would significantly strengthen the contribution

__presentation__: The presentation is generally clear and the results are written in a structured and understandable way. The mathematical development is organized and the derivations are presented with sufficient detail. That said, the paper would benefit from a stronger effort to translate the theoretical constructions into practical insights.

---

> ### Author Rebuttal · Authors · 2026-03-31
>
> We sincerely thank the reviewer for the time spent reviewing our submission.
>
> The most important practical advantage the new divergence provides relative to existing ones, such as KL, is that it can be state-space-aware, remaining well-defined in the variance-to-zero limit and thus providing information about how different the distributions' supports are. In contrast, KL uniformly gives the uninformative value $+\infty$.
> The submission already provides numerical experiments illustrating advantages in certain regimes: we show that for low noise (i.e., variance-to-zero), the WKL and KWKL yield informative policies, whereas the KL-regularized policy becomes the uninformative zero-policy.
>
> For comparison with other divergences, we refer to Remark 5.3. Alternative divergences induce control penalty terms beyond the LQR setting, which eliminates the possibility of closed-form Riccati-based solutions and necessitates more computationally intensive procedures.
>
> The main motivation of our canonical divergence in Def. 3.1 comes from the geodesic projection property. Say that we have a point $p$ which we want to project onto a manifold $M$ via a divergence $D(p \| q)$, that is, we search for a point $p^\ast \in M$ that satisfies $D(p \| p^\ast) = \inf_{q \in M} D(p\|q)$. Having such a projection $p^\ast$, it is natural to assume that the geodesic connecting $p$ with $p^\ast$ meets $M$ orthogonally.
> This property is satisfied for important examples: (a) for a Riemannian manifold where the divergence is given by half of the squared distance, the geodesic is given by the Levi-Civita connection, and the orthogonality is measured in terms of the Riemannian metric; (b) for the probability simplex, where the divergence is given by the Kullback-Leibler divergence, the geodesic is simply the mixture geodesic, and orthogonality is measured in terms of the Fisher-Rao metric; (c) for the probability simplex, where the divergence is given by the $\alpha$-divergence, the geodesic is the corresponding $\alpha$-geodesic, and orthogonality is measured in terms of the Fisher-Rao metric. In information geometry, the geodesic projection property singles out particular divergences, which were studied as canonical divergences in [AA15, FA21] and often turn out to be asymmetric in their arguments.
> This projection property does not hold for many other commonly used divergences.
>
> Inspired by the reviewer's comment, we have derived several properties of $D^G$, which we will include in our revision.
> * We have $D^G \ge 0$ and $D^G(\mu \mid \nu) = 0$ iff $\mu = \nu$. However, $D^{G}$ may take the value $+\infty$.
> * $D^G$ is asymmetric and does not fulfill the triangle inequality.
> * $D^{G}(\mu_1 \otimes \mu_2 \mid \nu_1 \otimes \nu_2) = D^{G}(\mu_1 \mid \nu_1) + D^G(\mu_2 \mid \nu_2)$ for the WKL and Gaussian distributions and, if the assumptions from Thm. 4.11 hold, also for the KWKL and Gaussian distributions.
> * On the set of elliptic distributions, WKL and KWKL have the same (lower semi-)continuity properties as the KL.
> * $D^{G}$ is invariant under isometries of the underlying space applied to both input measures simultaneously if the inertia operator is equivariant under the action of that isometry.
> Equivariance holds for any isometry $\Phi$ in the Fisher-Rao metric.
> We will add the proof that this is true for the Otto metric (with non-linear mobility if it acts by scalar multiplication) as well.
> For the Stein geometry with the generalized bilinear kernel, we have to restrict to isometries $\Phi(x) = Ox + b$ with $O^{\top} A O = A$.
>
> To demonstrate how this theoretical investigation can be converted into a modern RL-pipeline, we will revise the manuscript to include the following roadmap explicitly: \
> (i) the present paper establishes well-posedness and closed-form structure in an analytically tractable setting; \
> (ii) the proposed divergences are directly compatible with Gaussian policy parameterizations used in continuous-control RL and could potentially remedy the same KL blow-up pathology identified in KL-regularized RL from demonstrations [R21]; in particular, they can be integrated into trust-region style policy optimization as a natural replacement for the KL regularizer in continuous action spaces [S15]; \
> (iii) a natural next step is to develop sampling-based or local-approximation implementations in more general non-linear MDPs.
>
> [AA15] Ay, N., Amari, S. I. (2015). A novel approach to canonical divergences within information geometry. Entropy, 17(12), 8111-8129.
>
> [FA21] Felice, D., Ay, N. (2021). Towards a canonical divergence within information geometry. Information geometry, 4(1), 65-130.
>
> [R21] Rudner, T.G., Lu, C., Osborne, M.A., Gal, Y. and Teh, Y., 2021. On pathologies in KL-regularized reinforcement learning from expert demonstrations. NeurIPS 34, pp. 28376–28389.
>
> [S15] Schulman, J., Levine, S., Abbeel, P., Jordan, M. and Moritz, P., 2015. Trust Region Policy Optimization. ICML.

---

> > ### Author Rebuttal · Reviewer_hi3G · 2026-04-03
> >
> > I thank the authors for the response and addressing my concerns. The concern about comparison to other metrics is partially resolved.

---

### Official Review · Reviewer_SjXM · 2026-03-15

**Soundness:** 3
**Presentation:** 2
**Significance:** 2
**Originality:** 3
**Overall Recommendation:** 3
**Confidence:** 2

**Summary:**

The paper argues that the standard Kullback-Leibler (KL) regularization used in stochastic optimal control and reinforcement learning becomes degenerate in the limit of low-noise. Specifically, in the case of linear-Gaussian control, the quadratic control penalty scales with the inverse noise, which blows up as process noise becomes small.  The authors proposes replacing the KL regularization in KL-control problems by either a KL-Wasserstein or a Kalman-KL-Wasserstein divergence, avoiding the Fisher-Rao geometry and instead relying on Wasserstein-like geometries.

**Compliance With Llm Reviewing Policy:**

Affirmed.

**Final Justification:**

I appreciate the author’s rebuttal and their effort to clarify the paper. Although information geometry is not my primary area of expertise, I made an honest attempt to engage with the work in a constructive manner. However, the response has reinforced my initial skepticism. Given my limited ability to assess the depth and validity of the theoretical contributions, and my concerns regarding the decision-theoretic framing of the paper, I find it appropriate to lower my score. As it stands, the work's contribution appears to be rooted in information geometry rather than reinforcement learning or planning, and in my view it would be more appropriately evaluated within that context.

**Key Questions For Authors:**

* Why not just default to Wasserstein regularization? Why are WKL and KWKL geometries preferable?

* The authors write _"Furthermore, the gradient flows of simple energies in more complicated geometries are related to transformers."_ Could you please elaborate on that idea? I am not sure how it relates to the focus of the paper.

* Does the solution of the LQ problem converge to the solution of the Riccati solution of the deterministic system in the limit of vanishing noise?

**Limitations:**

The paper did not explicitly discuss limitations.

**Strengths And Weaknesses:**

**Disclaimer**

The extensive information geometry theory presented in the paper lies outside my main area of expertise and this is reflected in my low confidence score. Nevertheless, in good faith I will engage with the parts that I am familiar with and avoid down-scoring the paper at this stage. I will rely on the authors’ response, the remaining reviews, and the AC-reviewer discussion period to calibrate my final score.

**Clarity**

Despite the above disclaimer, my assessment is that the paper lacks clarity regarding both its motivation and contributions. The manuscript does not clearly articulate the main objective and fails to guide the reader through the background material and the main technical section.

The optimal control application appears somewhat disconnected from the main narrative. It reads more like an afterthought rather than the central focus of the work. The main emphasis of the paper seems to be developing general information-geometric alternatives to the KL.

 **Novelty**

Based on my understanding of the paper, the main technical contribution appears to be the generalization of Datar and Ay (2026) to the setting elliptical distributions and the introduction of the Kalman-Wasserstein KL, which would represent a substantial contribution.

**Major Comment**

I have considerable reservations with regard to the optimal control application. In divergence-regularized stochastic optimal control, regularization is typically formulated between controlled and uncontrolled path measures, rather than between conditional transition kernels. See for example the original path integral control formulation by Kappen and Theodorou. In section 5, however, the manuscript instead introduces a sum of stagewise penalties between conditional transition measures. For the standard KL divergence, such a decomposition is justified, as the path-space regularization leads to a sum of stagewise penalties. However, it is not clear to me that the proposed WKL and KWKL divergences would in fact factorize in the same way. In my opinion, this casts some doubt on this application.

**Minor Comments**

* Support mismatch is mentioned as a motivation in both the abstract and the introduction, but it is not developed further the evaluation section.

* Kullback-Leibler regularization in reinforcement learning appeared well before Trust-Region Policy Optimization by Schulman et al. See for example, Relative Entropy Policy Search by Peters et al.

* The paper argues that the divergence-based regularization is a principled approach to selecting the control penalty matrix. I find this argument unconvincing. The selection of the cost penalties is generally related to certain task specifications. While the divergence regularization does lead to an interesting structure of the penalty that includes the matrix $B$, augmenting the LQ cost with a divergence is still itself a heuristic. In practice, it seems likely that the divergence term would need to be balanced against the state penalty $Q$ via a tunable scaling factor.

---

> ### Author Rebuttal · Authors · 2026-03-31
>
> We thank the reviewer for recognizing our submission as a substantial contribution. We agree that the motivation and contributions need clarification and will revise Section 1.1 accordingly:
>
> We introduce G-dual geodesics and define $D^G$ as action integrals along them. This generalizes KL (recovered under Fisher–Rao geometry) and yields new cases such as KWKL.
> We show that G-dual geodesics coincide with G-metric gradient flows of a potential energy, linking our framework to optimization and mean-field limits of neural networks [CB20, HL26].
> We derive explicit formulas for $D^G$ between elliptic distributions under Wasserstein, linearized Wasserstein, Kalman–Wasserstein, and Stein geometries. For transport-based geometries, $D^G$ respects the state space geometry, unlike KL.
> Motivated by KL-regularized RL, we study model-based control with $D^G$ regularization and show that, in the low-noise regime, it yields well-defined optimal policies while retaining key KL properties.
>
> Regarding the third minor comment, we agree that the overall LQ objective still involves a task-dependent trade-off between state regulation and control regularization, and thus the state-cost matrix $Q$ remains problem-specific. Our claim is therefore not that the divergence removes the need for tuning altogether. Rather, it provides a principled way to induce the structure of the control penalty matrix. The regularizer determines an anisotropic control metric that depends on the noise covariance and the input matrix $B$, thereby fixing the relative penalty across the control directions. The divergence removes an ad hoc degree of freedom in the design of the control penalty itself. Put differently, the tuning of $Q$ reflects the task specification. In contrast, the divergence-based regularizer specifies how to measure the control effort in the action space, given the dynamics and noise structure. We will revise the manuscript to make this distinction explicit and to avoid overstating the claim.
>
> Regarding the third key question on the deterministic limit, Corollary 5.1 from the paper shows that for $\lambda=1$, WKL and KWKL converge to the deterministic LQR solution with $R=B^T B$, and for general $\lambda$, KWKL converges to the one with $R=\frac{1}{\lambda} B^T B$.
> In contrast, the vanishing-noise limit with KL corresponds to the classical expensive-control limit where the limiting controller depends on the open-loop spectral structure: if $\sqrt{\gamma} A$ is Schur stable, then the optimal policy converges to zero (see our Remark 5.2); otherwise, the limit is a nontrivial solution that reflects the unstable eigenvalues of $\sqrt{\gamma} A$ to their reciprocals inside the unit circle. This behavior is well known from the theory of LQR and is closely related to symmetric root-locus and spectral factorization results (see, e.g., Theorem 3.11 in [KS] for a continuous-time analogue).
>
> Regarding the major comment on factorization: our Section 5 follows the KL-control / linearly-solvable MDP (LMDP) framework [T06,T09,DT10,GRW14], where the regularization is defined directly at the one-step transition level, rather than derived from a trajectory-level divergence. In this setting, the stagewise KL is the primitive object that defines the control effort [2, Eq. (2)], and directly induces the quadratic control penalty with inverse noise scaling [2, Eq. (10)] that we analyze. Our contribution operates at exactly this level: we replace a stagewise divergence that becomes singular with alternatives that remain well-posed, while preserving tractability and interpretability.
> Importantly, this does not weaken the control formulation, as no path-space decomposition is required in LMDPs.
> Furthermore, we prove an additivity property (see response to hi3G) which may allow us to also address the path-integral formulation in the future.
> We will revise the manuscript to clearly distinguish the LMDP (stagewise) formulation from path-integral control, and to explicitly state that we do not claim any general path-space factorization property.
>
> [CB20] Chizat and Bach. Implicit bias of gradient descent for wide two-layer neural networks trained with the logistic loss (2020), COLT.
>
> [HL26] Hardion and Lavenant. Gradient Flows of Potential Energies in the Geometry of Sinkhorn Divergences. arXiv:2511.14278 (2025).
>
> [KS] Kwakernaak and Sivan, 1972. Linear Optimal Control Systems.
>
> [T06] Todorov. Linearly-solvable MDPs. NeurIPS (2006).
>
> [T09] Todorov. Efficient computation of optimal actions. PNAS (2009).
>
> [DT10] Dvijotham and Todorov. Inverse optimal control with LMDPs. ICML (2010).
>
> [GRW14] Guan, Raginsky, and Willett. Online MDPs with KL cost. IEEE TAC (2014).

---

> > ### Author Rebuttal · Reviewer_SjXM · 2026-04-01
> >
> > I appreciate the authors’ clarification and accept that the stagewise LMDP formulation is mathematically valid. My concern, however, is that this clarification weakens the motivation for the control application rather than resolving it. It reinforces my impression that the control perspective is somewhat of an afterthought.
> >
> > My concern is somewhat more philosophical. In my opinion, the appeal of the LMPD framework is that it defines a new tractable class of decision-making processes, and not that it offers any novel decision-theoretic interpretation. Much of the modern literature has moved on towards path-integral and information-theoretic formulations, where the KL term has a clear semantic meaning as the divergence between controlled and uncontrolled path measures. It is only because KL factorizes in the right way that these formulations can then be expressed within the LMDP structure.
> >
> > LMDP is a useful mathematical framework for describing a certain class of control problems, but by itself it does not confer decision-theoretic legitimacy on a particular stagewise penalty. It feels somewhat contrived to invoke the stagewise LMDP formulation in order to justify replacing the KL term without providing a clear control- or transport-theoretic motivation.

---

> > > ### Author Response · Authors · 2026-04-06
> > >
> > > We appreciate the reviewer’s thoughtful comment regarding the role of path-measure regularization in stochastic optimal control. We agree that the path-integral formulation (Kappen, Theodorou) provides the canonical decision-theoretic interpretation, in which the KL term represents the divergence between the controlled and uncontrolled trajectory measures. Our rebuttal clarified that Section 5 follows the LMDP formulation in which the stagewise divergence is taken as the primitive object.
> > >
> > > Motivated by your comment, a closer analysis of the factorization property revealed the following:
> > > Although we have an additivity property (see response to Reviewer hi3G) which allows us to factorize the objective function into a sum of stage-wise costs in the setting of memoryless systems, this does not work out for more general systems as you observed. This is because the key property facilitating the factorization is the chain rule decomposition which holds for the KL, but not for our divergences.
> > >
> > > Nevertheless, we agree it is important to understand whether our conclusions persist under the full path-space perspective.
> > > To address this, we derived an additional result analyzing the path-integral formulation directly in the linear–Gaussian finite-horizon setting.
> > > The following theorem demonstrates that the low-noise pathology of KL, as well as the well-posed limits of WKL and KW, also arise directly in the path-integral control formulation. Importantly, this result does not rely on any factorization or chain-rule of the divergence.
> > >
> > > We start again with
> > > $$x_{t+1}=Ax_t+Bu_t+w_t$$ with iid $w_t \sim N(0,\Sigma_w)$.
> > > Defining the stacked state, input, and noise sequences to be
> > > $$x = [x_1^\top,\ldots,x_T^\top]^\top, u = [u_0^\top,\ldots,u_{T-1}^\top]^\top, w = [w_0^\top,\ldots,w_{T-1}^\top]^\top$$ leads to controlled and uncontrolled trajectories
> > > $$x=\mathcal A x_0+\mathcal B_u u+\mathcal B_w w,$$ $$x_{uc}=\mathcal A x_0+\mathcal B_w w$$ for appropriately defined block-matrices $\mathcal A,\mathcal B_u, \mathcal B_w$.
> > > Following recent work [PHT2024], we now consider the finite-horizon objective
> > > $$V^\circ(u) = \mathbb{E}\left[\tfrac12 x^\top \mathcal Q x\right] + D^\circ \big(p(x)\|p(x_{uc})\big),$$ where $\circ\in\{\mathrm{KL},\mathrm{WKL},\mathrm{KW}\}$.
> > >
> > > ## Theorem (Path-space optimal control)
> > > For the above setup, the divergences between the Gaussian path measures reduce to quadratic penalties
> > > $$ D^\circ(p(x)\|p(x_{uc}))=\tfrac12 u^\top R_\circ u$$ with
> > > $$ R_{\mathrm{KL}}=\mathcal B_u^\top\Sigma^{-1}\mathcal B_u,R_{\mathrm{WKL}}=\mathcal B_u^\top \mathcal B_u,R_{\mathrm{KW}}=\mathcal B_u^\top(\Sigma+\lambda I)^{-1}\mathcal B_u,$$ where $\Sigma=\mathcal B_w(I \otimes \Sigma_w)\mathcal B_w^\top$.
> > > The optimal control sequence is $ u_\circ = -(R_\circ+\mathcal B_u^\top \mathcal Q\mathcal B_u)^{-1}\mathcal B_u^\top \mathcal Q \mathcal A x_0$.
> > > In particular $u_{\mathrm{WKL}}$ is independent of $\Sigma_w$.
> > > If $\Sigma_w=\rho I$ and $\lambda=1$, then
> > > $$
> > > \lim_{\rho\searrow0}u_{\mathrm{KW}}=u_{\mathrm{WKL}}, \quad
> > > \lim_{\rho\to\infty}\|u_{\mathrm{KW}}-u_{\mathrm{KL}}\|=0, \quad
> > > \lim_{\rho\searrow0}u_{\mathrm{KL}}=0 .
> > > \hspace{10cm} \square
> > > $$
> > >
> > > Interestingly, in the special case $\mathcal{B}_u=I$, the low-noise degeneracy of KL can already be inferred from [Theorem 2, PHT2024], which provides additional evidence that the phenomenon is intrinsic to the KL geometry.
> > >
> > > We will add this result and its proof in the revised manuscript to clarify that the control interpretation is not tied to the stagewise LMDP formulation and also holds within the standard path-space perspective.
> > > We hope that this analysis convinces the reviewer that our observations hold not only for the LMDP formulation but also for the path-integral formulation.
> > >
> > > We would like to emphasize that although we have presented the above extension in a finite-horizon setting involving a fixed initial state $x_0$, we are confident that this can be appropriately extended to our previous setting, the setting of an infinite horizon objective function with $x_0$ distributed according to some known Gaussian distribution. This is because the control penalty will still be inversely proportional to the variance leading to the same phenomenon.
> > >
> > > [PHT2024] Patil, A., Hanasusanto, G.A. and Tanaka, T., 2024. Discrete-time stochastic LQR via path integral control and its sample complexity analysis. IEEE Control Systems Letters, 8, pp.1595-1600.
> > >
> > > ### **Summary of both rebuttals:**
> > > To summarize both rebuttals, we believe all concerns are addressed as follows:
> > > (i) clarified objectives/contributions and committed to improving exposition and positioning
> > > (ii) acknowledged absence of a KL-type chain rule and that factorization holds only in the memoryless case
> > > (iii) **added a new path-space theorem showing KL degeneracy and well-posedness of WKL/KWKL without factorization**
> > > (iv) softened and clarified control-penalty claims
> > > (v) clarified deterministic limits ($\rho \rightarrow 0$).

---

### Decision · Program_Chairs · 2026-04-30

**Decision:**

Accept (regular)

**Comment:**

This paper begins by noting that standard KL-based regularized becomes degenerate in the low-noise limit, and proposed alternative is a family of divergence metrics that replace Fisher-Rao geodesics by natural geometric ones. Beyond some closed-form expression of such divergences, the paper gives the best regularized policy for linear systems, and how this changes for various distance measures.

This paper received recommendations to accept from three reviewers (including two clear accepts), and one weak reject. The reviewers appreciated the conceptual novelty and mathematical rigor. The weak reject evaluation raised a valid point regarding whether the WKL terms factorize into stagewise costs like KL. We agree that without it the control application looks like an afterthought (although still valid, just not very grounded). But the authors give a setting justifying this in the final rebuttal.

Given this, our recommendation is to accept the paper.